# Symbolic Recovery of PDEs from Measurement Data

**Erion Morina**                                                             *erion.morina@uni-graz.at*
*IDea_Lab - The Interdisciplinary Digital Lab at the University of Graz, Austria*

**Philipp Scholl**                                              *philipp.scholl@aleph-alpha-research.com*
*Aleph Alpha Research, Germany*

**Martin Holler**[*]                                                         *martin.holler@uni-graz.at*
*IDea_Lab - The Interdisciplinary Digital Lab at the University of Graz, Austria*

**Reviewed on OpenReview:** *https://openreview.net/forum?id=TbHfgo1OW3*

## Abstract

Models based on partial differential equations (PDEs) are powerful for describing a wide range of complex phenomena in the natural sciences. Accurately identifying the PDE model, which represents the underlying physical law, is essential for a proper understanding of the problem. This reconstruction typically relies on indirect and noisy measurements of the system's state and, without specifically tailored methods, rarely yields symbolic expressions, thereby limiting interpretability.

In this work, we address this limitation by considering neural network architectures based on rational functions for the symbolic representation of physical laws. These networks combine the approximation power of rational functions with the flexibility to represent arithmetic operations, and generalize *ParFam* and *EQL*-type architectures used in symbolic regression for physical law learning. We further establish regularity results for these symbolic networks.

Our main contribution is a reconstruction result showing that, if there exists an admissible physical law that is expressible within the symbolic network architecture, then in the limit of noiseless and complete measurements, symbolic networks recover a physical law within the PDE model that is representable by the architecture. Moreover, the recovered law corresponds to a regularization-minimizing parameterization, promoting interpretability and sparsity in case of $L^1$-regularization. Under an additional identifiability condition, the unique true physical law is recovered.

These reconstruction and regularity results are derived at the continuous level prior to discretization due to a formulation in function space. Empirical results using the ParFam architecture are consistent with the theoretical findings and suggest the feasibility of reconstructing interpretable physical laws in practice.

*Keywords:* Physical law, model learning, symbolic recovery, neural networks, rational functions, inverse problems

*MSC Codes:* 35R30, 93B30, 68Q32, 65M32, 41A20

---

[*]MH is also a member of NAWI Graz (www.nawigraz.at) and BioTechMed Graz (biotechmedgraz.at).

# 1 Introduction

Many complex relationships in the natural sciences are governed by systems that evolve in space and time, driven by rates of change. The underlying physical laws can be effectively modeled using partial differential equations (PDEs), which provide a powerful mathematical framework for describing such effects while enabling the analysis and prediction of system behavior. Examples include modeling fluid dynamics with the Navier-Stokes equations Batchelor (2000), behavior of quantum systems with Schrödinger's equation Griffiths & Schroeter (2018), biological pattern formation through reaction-diffusion equations Turing (1952), climate dynamics and weather prediction Vallis (2017), and biological processes such as population dynamics or transport mechanisms Murray (2002; 2003); Perthame (2015). Despite their versatility, a central challenge in PDE-based modeling lies in identifying the precise form of the governing equations, which is often unclear or imprecise. This uncertainty stems from multiple factors, such as insufficient understanding of the underlying mechanisms or oversimplifications during the abstraction process that prevent accurate representation of reality. Moreover, the data available in practice are often indirect, incomplete, and noisy, which further complicates the reconstruction of the governing equations.

This reconstruction task can be naturally cast as an inverse problem. Classical inverse problems and parameter identification methods (e.g., Banks & Kunisch (1989); Engl et al. (1996)) focus primarily on estimating unknown coefficients within a prescribed PDE structure, while recent scientific machine learning approaches for PDEs (Kovachki et al. (2024); Boullé & Townsend (2024)) learn more flexible, data-driven representations of the underlying dynamics. Yet, the symbolic, human-interpretable recovery of PDE laws together with rigorous identifiability guarantees in realistic measurement settings remains less explored. Beyond predictive accuracy, many scientific applications require interpretable models, where the governing law is expressed in compact, symbolic form, facilitating physical understanding, verification, and analysis (e.g., Brunton et al. (2016); Rudy et al. (2017)). Symbolic regression methods La Cava et al. (2021), e.g., based on symbolic neural network architectures (Lampert & Martius (2017); Sahoo et al. (2018); Scholl et al. (2025)) aim at discovering such human-readable laws. Since physical law learning is inherently an identification problem, it is crucial that the learned equations are uniquely determined by the data. Structural and parameter identifiability have been extensively studied for dynamical systems and PDE inverse problems (see, e.g., Bellman & Åström (1970); Isakov (1993) for early contributions). In the context of symbolic recovery of differential equations, recent work analyzes uniqueness and robustness of the recovery (Scholl et al. (2023a;b); Hauger et al. (2025)). However, analytic reconstruction guarantees for symbolic recovery of PDEs from measurement data at function level remain open. In this work, we address this gap by developing a function-space identification framework for symbolic model learning and establishing corresponding reconstruction and regularity results.

## 1.1 Scope and contributions

Our focus is the joint recovery of an unknown physical law $f$ and state $u$ satisfying the PDE

$$\partial_t u(t, x) = f(t, u(t, x)), \qquad (t, x) \in (0, T) \times \Omega,$$

on a given spatio-temporal domain, from indirect, noisy and possibly incomplete measurements of the state such that $f$ is given in a concise symbolic form. The aim is not merely to fit a physical law $f$ numerically, but to recover it as a human-interpretable expression that is verifiable

by domain experts. To achieve this, we approximate $f$ by a family of parameterized models designed for symbolic expressiveness and interpretability. Concretely, we employ *symbolic networks*, a neural-network based architecture whose trainable components are rational functions (fractions of polynomials with trainable coefficients) and whose activations are user-defined base functions (such as trigonometric or exponential functions). Architectures commonly used for neural-network based symbolic regression, including *EQL*-type designs Lampert & Martius (2017); Sahoo et al. (2018) and *ParFam* Scholl et al. (2025), inspired our choice of symbolic neural networks and are subsumed as special cases within our framework. The design outlined above is crucial since rational functions naturally encode arithmetic operations, such as products and divisions, while the base functions provide the vocabulary for non-algebraic relationships. Together, they yield a modeling class that can represent compact formulas and is supported by strong approximation capabilities grounded in those of rational functions (see Boullé et al. (2020); Holler & Morina (2025a); Telgarsky (2017)). We integrate symbolic networks into the model learning framework proposed in Holler & Morina (2024a), that recovers the state and the physical law simultaneously from data. This allows to incorporate measurement operators and noise models directly into the learning scheme. Within this formulation, sparsity-promoting regularization on the model parameters is used to enhance interpretability of the resulting physical law. As a result, among all admissible models consistent with the data, a concise symbolic law within the chosen architecture is preferentially selected.

Our main theoretical contribution is a reconstruction result tailored to this symbolic setting. Specifically, we show if there exists an admissible physical law expressible within a user-defined symbolic network architecture, then the proposed model learning approach recovers a physical law and state consistent with the PDE model in the limit of complete measurements. The resulting parameterization of the physical law is regularization minimizing, which is particularly interesting if the regularization scheme is designed to promote sparsity in the parameterization. Furthermore, under an identifiability condition, the symbolic networks recover the unique ground truth physical law.

When the physical law is not representable by a fixed architecture, techniques developed in Holler & Morina (2024a) can be employed, but fall outside the scope of this work. Beyond theory, we provide a numerical validation using ParFam, demonstrating the practical feasibility of reconstructing interpretable laws from indirect data. All numerical experiments of this work can be reproduced with the publicly available source code[1].

## 1.2 Novelty

What distinguishes this work from prior PDE identification frameworks and neural-network based symbolic regression architectures such as ParFam is a function-space analysis specifically tailored to symbolic networks. While existing literature, such as Holler & Morina (2024a), provides results on PDE-based model identification formulated at the level of functions, it is not guaranteed that the learned models admit an interpretable representation. To address this limitation, we employ symbolic networks to represent the unknown physical law. Importantly, our approach is not a straightforward adaptation of existing all-at-once frameworks to a symbolic parameterization. Although the framework in Holler & Morina (2024a) is stated in general terms, many of its abstract assumptions are nontrivial to verify for symbolic neural networks, and it is not evident a priori whether the theory applies in this setting. In particular, the required regularity and mapping properties have not been established for architectures such as ParFam or EQL-type networks, nor are

---

[1]See https://github.com/Philipp238/unique_sym.

they covered by existing symbolic regression theory. A central methodological feature of this work is the modeling of the dynamics directly in function space prior to discretization. This preserves the continuous structure of the PDE and allows us to establish reconstruction guarantees at the analytical level, independent of any numerical approximation. Building on this perspective, we develop an analysis-guided framework for reconstructibility in symbolic PDE recovery, supported by both theoretical guarantees and first numerical evidence of practical feasibility.

Taken together, this work establishes an analysis-driven framework for the symbolic recovery of physical laws from measurement data by combining a function-space formulation of the learning problem and new regularity results for symbolic networks such as ParFam and EQL-type architectures. For that, we (i) parameterize physical laws by symbolic networks based on rational functions in an all-at-once formulation that incorporates incomplete measurement operators and noise; (ii) analyze these architectures as operators on function spaces and establish regularity properties required for the learning problem; and (iii) derive reconstructibility results for neural-network based symbolic model learning that have not previously been available.

### 1.3 Organization

Section 2 reviews related work. Section 3 introduces symbolic networks based on rational functions, as detailed in Subsection 3.1. The network design is outlined in Subsection 3.2, and approximation properties are established in Subsection 3.3. Subsection 3.4 discusses the extension to function space, including regularity properties of symbolic networks. The main results for symbolic model learning are presented in Section 4, including the analytical framework in Subsection 4.1, the core reconstruction results in Subsection 4.2, and possible extensions in Subsection 4.3. Numerical experiments are provided in Section 5. The detailed proofs of the regularity result from Subsection 3.4 can be found in Appendix A, and those for Section 4 in Appendix B. Appendix C verifies the regularity properties of sampling operators utilized in the numerical experiments.

## 2 Related work

This section reviews related work on learning PDE-based models from data, with a focus on inverse problems, interpretable and symbolic model discovery, and identifiability of physical laws. These directions are most closely related to the setting and goals of this paper.

**Inverse problems and PDE model identification.** The reconstruction of physical laws in PDE-based models from measurement data can be formulated as an inverse problem. Applications range from estimating single parameters, through identifying structural components that augment approximate models, to discovering the full governing physics that best explain observed dynamics. Approaches span classical techniques, purely data-driven model learning, and hybrid methods.

In parameter identification the physical model structure is assumed to be known and only the physical parameters are learned from data. Early contributions include Acar (1993), Alessandrini (1986), and Knowles (2001) on elliptic equations. Related works on reconstructing nonlinear heat conduction laws from boundary measurements, together with stability estimates, include Cannon & DuChateau (1980), Egger et al. (2014), and Rösch (1996). An overview on aspects of parameter identification and beyond is provided by the foundational reference works Banks & Kunisch (1989) on parameter estimation and control for PDE-governed systems, Klibanov (2013) on different as-

pects of coefficient inverse problems, Engl et al. (1996) on deterministic regularization theory for ill-posedness, and Kaipio & Somersalo (2005) on a Bayesian framework for inverse problems.

Beyond coefficient estimation, model identification infers unknown structural components of PDEs, e.g., full functional terms, from data, including recovery of sources in reaction-diffusion systems DuChateau & Rundell (1985); Kaltenbacher & Rundell (2020a; 2025), joint reconstruction of conductivity and reaction kinetics Kaltenbacher & Rundell (2020b), determination of nonlinearities in semilinear equations Feizmohammadi et al. (2024); Kian (2023); Kian & Uhlmann (2023), inverse source problems for hyperbolic dynamics Yamamoto (1995) and hidden reaction law discovery Ngoc Nguyen (2025). See also Bukhgeim & Uhlmann (2002); Jiang et al. (2017); Nachman (1996).

Efficiency is a central consideration, as measurement data are often high dimensional. For that reason, dimensionality reduction is a natural preprocessing step prior to inferring physical laws. Representative approaches include projection-based model reduction for parametric PDE systems Benner et al. (2015) and data-driven dynamic mode decomposition (DMD), which extracts dominant spatio-temporal modes from measurements Schmid (2010). Another central consideration is uncertainty. The framework Stuart (2010) for Bayesian inverse problems on function spaces enables probabilistic inference with principled uncertainty quantification, facilitating the identification of governing parameters and laws from partial, noisy data.

**Scientific machine learning for PDEs.** In recent years, scientific machine learning has shown strong potential by linking data-driven methods with classical models. For an overview of learning PDE-based models from data, see the comprehensive reviews Azizzadenesheli et al. (2024); Boullé & Townsend (2024); Brunton & Kutz (2024); De Ryck & Mishra (2024); Nganyu Tanyu et al. (2023). A selection of methods include DeepONets Lu et al. (2021), Fourier Neural Operators Li et al. (2020), DeepGreen Gin et al. (2021) and model reduction with neural operator methods Bhattacharya et al. (2021). Hybrid methods have also proven effective in optimal control, including learning-informed optimal control Dong et al. (2022; 2024; 2025), nonlinearity identification in the monodomain model Court & Kunisch (2022) and identification of semilinear PDEs Christof & Kowalczyk (2024). In the setting of learning-informed PDE-constrained optimization, we refer to Riedl et al. (2025); Sirignano et al. (2023). These contributions formulate and analyze PDE-constrained optimization problems augmented with neural networks, provide adjoint-based optimization frameworks, and establish global convergence guarantees for neural PDEs and neural-network-enhanced PDE models. On learning-informed identification of PDE nonlinearities, we refer further to Aarset et al. (2023) for well-posedness of the learning problem and Holler & Morina (2024a) for uniqueness of reconstructing the physical law, along with the references therein. These contributions employ an all-at-once formulation, justified in Kaltenbacher (2016); Kaltenbacher & Nguyen (2022), that facilitates the direct use of measurements and avoids explicit parameter-to-state maps. A growing body of recent research investigates how embedding problem-specific conservation laws into machine-learning frameworks can aid in discovering governing physical laws. We refer to the related works discussed in Holler & Morina (2025b) for a recent overview.

**Learning interpretable models from data.** A key objective in learning scientific models from data is to obtain representations that are not only accurate but also interpretable, compact, and aligned with physical principles. In many applications, one aims to identify physical laws that are as simple as possible while still adequately capturing the data (see, e.g., Angelis et al. (2023); Quade et al. (2016)). Beyond interpretability, symbolic models are advantageous over black-box alterna-

tives for capturing fundamental physical principles and are more robust in terms of generalization Rudy et al. (2017). Interpretable models also offer reduced computational complexity for real-time applications Schmidt & Lipson (2009). Finally, symbolic expressions offer the additional advantage of being easier to verify for physical consistency, while analytical properties (such as stability and equilibria) are easier to extract Brunton et al. (2016). For background of various methods on symbolic regression, see La Cava et al. (2021). This includes genetic programming approaches Augusto & Barbosa (2000); Cranmer (2023); Schmidt & Lipson (2009; 2010), reinforcement learning formulations Mundhenk et al. (2021); Petersen et al. (2021); Sun et al. (2023), neural-network based methods Heim et al. (2020); Lampert & Martius (2017); Sahoo et al. (2018); Scholl et al. (2025), and transformer-driven expression models Biggio et al. (2021); Holt et al. (2023); Kamienny et al. (2022). Within symbolic recovery of dynamical systems, substantial progress has been made on identifying parsimonious laws for nonlinear dynamics. The SINDy framework Brunton et al. (2016) introduced sparse regression to discover governing ODEs, and PDEs in Rudy et al. (2017). Physical structure can be promoted through priors and constraints, including symmetry-based inductive biases Udrescu & Tegmark (2020) and the incorporation of established scientific laws Cornelio et al. (2023), improving physical consistency, and data efficiency.

**Identifiability of physical laws.** In physical law learning, fundamentally an identification problem, it is crucial that the learned equations and parameters are uniquely determined by the data, ensuring the model captures the true underlying mechanism rather than one of many indistinguishable alternatives. The works Bellman & Åström (1970); Cobelli & DiStefano (1980); DiStefano & Cobelli (1980); Miao et al. (2011) establish the theoretical foundations of structural and parameter identifiability in dynamical systems. They rigorously define conditions for unique parameter inference from input-output data and analyze ambiguities that arise from observability and model structure. See for example Isakov (1993) investigating conditions under which inverse problems for semilinear parabolic equations admit unique solutions and Kaltenbacher & Rundell (2021) on identifiability and constructive recovery of a nonlinear diffusion coefficient in parabolic equations. In the context of symbolic recovery of differential equations, identifiability, specifically, the classification of uniqueness, has been addressed by Hauger et al. (2025); Scholl et al. (2023a;b) both for specific classes of physical laws (e.g., linear, algebraic) and for robust symbolic recovery of differential equations. More recently, Casolo et al. (2025) study the structural identifiability of sparse linear ODEs and show that, unlike in the dense case, non-identifiability occurs with positive probability under realistic sparsity assumptions. An interesting recent result by Shumaylov et al. (2025) shows that a system's governing equations are, in general, discoverable from data only when the system exhibits chaotic behavior. Finally, we also refer to Holler & Morina (2024a), addressed above, which discusses an analysis-based guideline for expecting unique reconstructibility of practical physical law learning setups in the limit of full measurements in a variational formulation.

## 3 Symbolic networks

Neural network-based symbolic regression provides an alternative to traditional black-box models by aiming to discover human-interpretable equations that describe data (e.g., originating from physical systems). Unlike genetic programming approaches, which rely on evolutionary algorithms to combine mathematical symbols, these methods exploit the potential of optimization techniques for neural networks (e.g., gradient-based methods). In addition, neural network-based approaches enable the construction of concise symbolic expressions and, when properly trained, generalize well

to unseen data. We focus on *symbolic networks*, a generic class of architectures which we motivate by the parameterized feed-forward neural networks proposed in Lampert & Martius (2017); Sahoo et al. (2018); Scholl et al. (2025) in the context of symbolic regression. These networks utilize user-defined base functions (e.g., trigonometric, exponential) as activation functions, which may vary across different layers. Transformations between layers are parameterized functions which are *simpler* than the activations and can range from affine linear transformations (as used in Lampert & Martius (2017); Sahoo et al. (2018)) to polynomial and rational transformations (see Scholl et al. (2025)). The symbolic networks investigated here specifically employ certain rational functions as transformations between layers, i.e., quotients of polynomials, encompassing polynomial and linear functions as special cases. Rational transformations can represent products and divisions, which are essential arithmetic building blocks for symbolic expressions. Additionally, they exhibit strong approximation capabilities, which will be discussed in more detail later.

### 3.1 Rational functions

From both a theoretical and practical perspective, it is advantageous to restrict transformations between layers to rational functions that do not exhibit poles. Rational functions with poles (e.g., outside the domain of the PDE model) are beyond the scope of this work, as the focus here is to enable a more universal and comprehensive analysis. This restriction guarantees essential regularity properties of symbolic networks, as proven in Subsection 3.4, while also simplifying the training process in practice. Furthermore, as we will discuss in Subsection 3.3, this limitation does not affect the approximation capabilities of rational functions.

Before introducing rational functions in the context described above, we first recall some definitions related to multivariate polynomials. For $m, n \in \mathbb{N}$ the degree of a multivariate polynomial $\pi : \mathbb{R}^n \to \mathbb{R}$ of the form

$$\mathbb{R}^n \ni x \mapsto \sum_{k_1,\ldots,k_n=0}^{m} a_{k_1,\ldots,k_n} \prod_{l=1}^{n} x_l^{k_l}$$

is defined by $\deg(\pi) := \max(k_1 + \cdots + k_n : a_{k_1,\ldots,k_n} \neq 0)$. Note that a polynomial of degree $d$ in $n$ variables involves at most $\binom{n+d}{d} = \frac{(n+d)!}{d!n!}$ non-zero coefficients. Additionally, note that two multivariate polynomials $p, q$ are *coprime* if they do not attain a nontrivial common polynomial divisor. Building on these definitions, we can now introduce rational functions and define an associated notion of degree.

**Definition 1** (Rational and base function). *For $n \in \mathbb{N}$, a function $r : \mathbb{R}^n \to \mathbb{R}$ is called rational if there exist coprime multivariate polynomials $p, q : \mathbb{R}^n \to \mathbb{R}$ with $q(x) > 0$ for all $x \in \mathbb{R}^n$, such that $r(x) = \frac{p(x)}{q(x)}$ for $x \in \mathbb{R}^n$. The degree of $r$ is given by $\deg(r) := \max(\deg(p), \deg(q))$ if $p \neq 0$ and zero otherwise. For $n, m \in \mathbb{N}$ a function $r : \mathbb{R}^n \to \mathbb{R}^m$ with $r(x) = (r_i(x))_{i=1}^m$ for $x \in \mathbb{R}^n$ is called rational if $r_i$ is rational for $1 \leq i \leq m$ and the degree is given by $\deg(r) := \max_{1 \leq i \leq m}(\deg(r_i))$.*

*A continuous function $\sigma : \mathbb{R}^n \to \mathbb{R}^m$ is called base function if it is not rational.*

To simplify the parameterization of denominator polynomials in the representation of rational functions (Definition 1), we introduce the set of coefficients for polynomials of a given degree that are positive over the real numbers.

**Definition 2** (Positive polynomials). *For $d, n \in \mathbb{N}$ let $Q(d, n) \subset \mathbb{R}^{\binom{n+d}{d}}$ be the set of coefficients parameterizing positive polynomials of degree $\leq d$ in $n$ variables,*

$$Q(d, n) = \{a \in \mathbb{R}^{\binom{n+d}{d}} : q(a, x) := \sum_{0 \leq k_1 + \cdots + k_n \leq d} a_{k_1, \ldots, k_n} \prod_{l=1}^{n} x_l^{k_l} > 0 \text{ for all } x \in \mathbb{R}^n\}.$$

To characterize the set of coefficients $Q(d, n)$ more explicitly, it is crucial to examine the geometric properties of polynomial roots, which form a rich area of research. However, as this is not the focus of the present work, we limit our discussion to classical one-dimensional results presented in (Jacobson, 2009, Chapter 5), specifically *Sturm's theorem* (Jacobson, 2009, Theorem 5.5) and *Tarski's theorem* (Jacobson, 2009, Theorem 5.9). By Sturm's theorem, the number of roots of a polynomial within a given interval is determined by the difference in the number of sign changes in the *standard sequence* evaluated at the endpoints of the interval. For $n = 1$, this establishes an implicit characterization of $Q(d, n)$ insofar as the number of variations in sign of the standard sequence remains constant for every real number. While an explicit characterization of $Q(d, n)$ is likely infeasible, it is typically sufficient to work with the following subset: Denominator polynomials consisting solely of monomial terms with even degrees, nonnegative coefficients, and a positive constant term. Such polynomials are positive over the real line and hold practical significance, as discussed next.

In Boullé et al. (2020), it is demonstrated that suitably regular functions can be uniformly approximated by rational functions on compact sets. The rational functions employed are *Zolotarev* sign-type functions, whose denominators, as noted previously, exclusively consist of monomials with even degrees. Similarly, this applies to the *Newman* sign-type functions used in Holler & Morina (2025a), which establish a first-order universal approximation result with rational functions. From a practical perspective, this eliminates the need for explicitly characterizing $Q(d, n)$, allowing one to instead focus on working with this proper subset without losing approximation properties.

### 3.2 Network design

The symbolic networks introduced in this work are inspired by the feed-forward neural networks proposed in Lampert & Martius (2017); Sahoo et al. (2018); Scholl et al. (2025). They extend the (shallow) ParFam architecture presented in Scholl et al. (2025) by incorporating multiple layers, and generalize the EQL architectures in Lampert & Martius (2017); Sahoo et al. (2018) by utilizing rational functions, according to Definition 1, as transformations between layers. The proposed symbolic network architecture is formally defined as follows.

**Architecture.** The depth of the network is denoted by $L \in \mathbb{N}$ and the widths by $(n_i^\sigma)_{i=0}^{L}, (n_i^r)_{i=1}^{L} \subset \mathbb{N}$. With this, we consider rational functions

$$r_i : \mathbb{R}^{n_{i-1}^\sigma} \to \mathbb{R}^{n_i^r}$$

of degree $d_i \in \mathbb{N}$ for $1 \leq i \leq L$ as transformations between the intermediate layers. The activation functions are given by fixed multivalued $n_i^r$-ary base functions

$$\sigma_i : \mathbb{R}^{n_i^r} \to \mathbb{R}^{n_i^\sigma}$$

for $1 \leq i \leq L$. Recall Definition 1 that *base function* cannot be expressed by a rational function. Note further that the presented setup also covers single unary base functions stacked within a

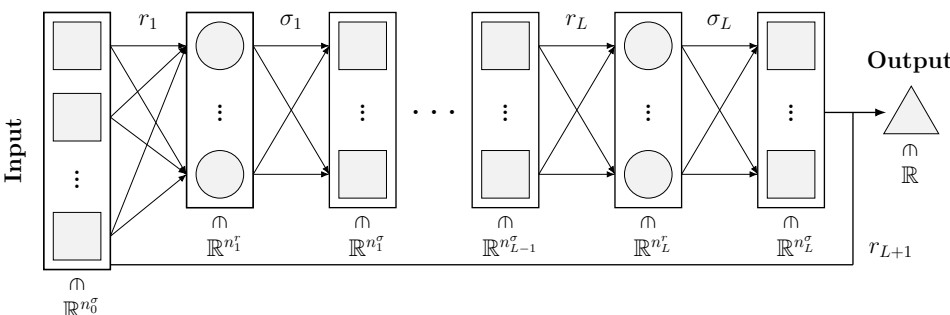

Figure 1: Scheme of symbolic network $\mathfrak{S}_\sigma$

multi-dimensional array by choosing $n_i^r = n_i^\sigma$ for $1 \leq i \leq L$. This is the case, for example, in the shallow ParFam architecture in Scholl et al. (2025). Finally, the transformation of the output layer is given by a rational function, including a skip connection, as described in Scholl et al. (2025), taking the form

$$r_{L+1} : \ \mathbb{R}^{n_L^\sigma + n_0^\sigma} \to \mathbb{R}.$$

The proposed symbolic network architecture is now generally given by

$$\begin{aligned} \mathfrak{S}_\sigma &: \mathbb{R}^{n_0^\sigma} \to \mathbb{R} \\ x &\mapsto r_{L+1}((\sigma_L \circ r_L \circ \cdots \circ \sigma_1 \circ r_1)(x), x). \end{aligned} \tag{1}$$

A schematic illustration of $\mathfrak{S}_\sigma$ is provided in Figure 1. We denote parameterizations of $\mathfrak{S}_\sigma$ by $\mathfrak{S}_\sigma^\theta$ for $\theta \in \Theta$ for a suitable parameter set $\Theta$, which will be explained in detail below. The complexity of the network $\mathfrak{S}_\sigma^\theta$ is controlled by the choice of the depth $L \in \mathbb{N}$, the layer widths $(n_i^\sigma)_{i=0}^L, (n_i^r)_{i=1}^L \subset \mathbb{N}$, and the degrees $(d_i)_{i=1}^{L+1} \subset \mathbb{N}$ of the underlying rational functions.

**Parameterization.** The trainable parameters $\theta = (\theta_i)_{i=1}^{L+1}$ of the network in (1), denoted in parameterized form by $\mathfrak{S}_\sigma^\theta$, consist of the coefficients $\theta_i$ of the polynomials that define the rational functions $r_i$ for $1 \leq i \leq L+1$ (see Definition 1). In practice, the parameterization is restricted to ensure both closedness of the parameter set and numerical stability during parameter training. Notably, the set $Q(d, n)$, from which the coefficients of the denominator polynomials of the rational functions $r_i$ are drawn (see Definition 2), is not closed. To ensure closedness of $\Theta$, we require a restriction of the denominator polynomials as follows. We define for $d, n \in \mathbb{N}$ and $\epsilon > 0$, for $q : \mathbb{R}^{\binom{n+d}{d}} \times \mathbb{R}^n \to \mathbb{R}$ given as in Definition 2, the set

$$Q^\epsilon(d, n) := \left\{ b \in \mathbb{R}^{\binom{n+d}{d}} : q(b, x) \geq \epsilon \quad \text{for all } x \in \mathbb{R}^n \right\}.$$

Closedness of $Q^\epsilon(d, n)$ follows from continuity of $q$, since

$$Q^\epsilon(d, n) = \bigcap_{x \in \mathbb{R}^n} [q(\cdot, x)]^{-1}([\epsilon, \infty)).$$

Another issue that arises alongside closedness is that rational functions, as introduced in Definition 1, are scaling-invariant, i.e., the numerator and denominator polynomials can be scaled by a positive

factor without changing the rational function itself. To avoid scaling invariance, we normalize the coefficients of the denominator polynomials with respect to the standard Euclidean norm $\|\cdot\|_2$, i.e.,

$$\bar{Q}^\epsilon(d, n) = \{b \in Q^\epsilon(d, n) : \|b\|_2 = 1\}.$$

Note that this strategy is also applied in ParFam Scholl et al. (2025). Taking this normalization into account, along with the earlier considerations regarding the design of the neural network, we can now define the closed parameter set $\Theta$, with $m_i = n_i^\sigma$ for $0 \le i \le L-1$ and $m_L = n_L^\sigma + n_0^\sigma$, for a small $\epsilon > 0$ by

$$\Theta = \bigotimes_{i=1}^{L+1} \left( \otimes_{k=1}^{n_i^r} \mathbb{R}^{\binom{m_{i-1}+d_i}{d_i}} \times \otimes_{k=1}^{n_i^r} \bar{Q}^\epsilon(d_i, m_{i-1}) \right),$$

where, for $i = 1, \ldots, L+1$, the first component $\otimes_{k=1}^{n_i^r} \mathbb{R}^{\binom{m_{i-1}+d_i}{d_i}}$ captures the coefficients of the numerator polynomials, and the second component $\otimes_{k=1}^{n_i^r} \bar{Q}^\epsilon(d_i, m_{i-1})$ captures the coefficients of the denominator polynomials.

### 3.3 Universal approximation

This subsection provides an overview of the universal approximation properties of symbolic networks $\mathfrak{S}_\sigma$ as introduced in (1). The ability of $\mathfrak{S}_\sigma$ to approximate functions of suitable regularity is fundamentally connected to the approximation capabilities of rational functions, which form the trainable components of the network. Reducing the question of approximability to rational functions is particularly significant, as it allows deriving results that are qualitatively independent of the specific choice of the activation (base) functions $\sigma$ within $\mathfrak{S}_\sigma$.

In the context of rational functions, we highlight the seminal work Telgarsky (2017), which investigates the reciprocal approximability of certain ReLU-activated neural networks and rational functions. Following this, related results in Boullé et al. (2020) focus on rational functions defined as compositions of low-degree rational functions, showing that comparatively simpler rational functions retain significant approximation capabilities. Moreover, for the uniform first-order approximation of smooth functions by rational functions, we refer to recent advances in Holler & Morina (2025a). Further classical results on rational approximation theory can be found in Petrushev & Popov (1988).

To conclude, we briefly discuss some key approximation properties of the symbolic network (1) as studied in Holler & Morina (2025a). It is shown that functions possessing sufficient regularity can be uniformly approximated up to their first-order derivatives by rational functions, as introduced in Definition 1, with positive denominator polynomials. This result can be extended directly to $\mathfrak{S}_\sigma$, given its architecture, which incorporates a rational skip connection. Finally, we provide a restricted formulation of the results in (Holler & Morina, 2025a, Lemma 16, Corollary 18), which discusses universal approximation properties of the network in (1) in the case where the layers are of equal size.

**Proposition 3.** *Consider symbolic networks in (1) with $d = n_i^r = n_i^\sigma$ and Lipschitz continuous activations $\sigma_i : \mathbb{R}^d \to \mathbb{R}^d$, such that $\sigma_i^{-1} \in \mathcal{C}^3([0,1]^d)$ for $1 \le i \le L$. Then, for every $f \in \mathcal{C}^3([0,1]^d)$ there exists a sequence of networks $(\mathfrak{S}_\sigma^m)_m$, with*

$$\lim_{m \to \infty} \|\mathfrak{S}_\sigma^m - f\|_{L^\infty([0,1]^d)} = 0.$$

*If $\nabla\sigma_i$ is Lipschitz continuous for $1 \leq i \leq L$, then $(\mathfrak{S}_\sigma^m)_m$ can be chosen such that*

$$\lim_{m\to\infty} \|\mathfrak{S}_\sigma^m - f\|_{L^\infty([0,1]^d)} + \|\nabla\mathfrak{S}_\sigma^m - \nabla f\|_{L^\infty([0,1]^d)} = 0.$$

*Proof.* See (Holler & Morina, 2025a, Lemma 16, Corollary 18). □

### 3.4 Extension to function space

The data arising from physical problems, while often measured discretely, is inherently functional in nature, representing some observable state variable (e.g., temperature or concentration) and potentially its higher-order derivatives. This motivates studying symbolic networks $\mathfrak{S}_\sigma$ within a function space formulation. Modeling the dynamics directly in function space preserves the inherent structure of the continuous problem and ensures consistency and stability prior to numerical discretization (see Hinze et al. (2009); Kaipio & Somersalo (2005); Stuart (2010)). To rigorously justify the application of symbolic networks to functional data, it is necessary to analyze the behavior of their architecture within the context of function space analysis.

#### 3.4.1 Framework

We follow the setup of Holler & Morina (2024a) and consider for fixed $T > 0$ the state $u \in \mathcal{V}$ as $u : (0,T) \to V$ with $\mathcal{V}$ the dynamic extension of $V$, the static state space consisting of functions $v : \Omega \to \mathbb{R}$. Here, $\Omega \subset \mathbb{R}^d$ denotes a bounded Lipschitz domain for some $d \in \mathbb{R}$. For $\kappa \in \mathbb{N}_0$, the order of differentiation, $\mathcal{J}_\kappa$ is a derivative operator

$$\begin{aligned} \mathcal{J}_\kappa : V &\to \otimes_{k=0}^\kappa V_k^\times \\ v &\mapsto (v, J^1 v, \ldots, J^\kappa v) \end{aligned} \tag{2}$$

and Jacobian mappings $J^k : V \to V_k^\times, v \mapsto (D^\beta v)_{|\beta|=k}$, for $1 \leq k \leq \kappa$. The spaces $V_k$ are such that $V \hookrightarrow V_k$ and $D^\beta v \in V_k$ for $1 \leq k = |\beta| = \beta_1 + \cdots + \beta_d \leq \kappa$ and $\beta \in \mathbb{N}_0^d$. With $V_0 := V$ the spaces $V_k^\times$ are further defined as $V_k^\times = \otimes_{i=1}^{p_k} V_k$, where $p_k = \binom{d+k-1}{k}$ for $0 \leq k \leq \kappa$. For $W$ the static image space, with $\mathcal{W}$ its dynamic extension, some unknown physical law $f$ is given as the Nemytskii operator of

$$f : \ (0,T) \times \otimes_{k=0}^\kappa V_k^\times \to W$$

where the latter is obtained by extending $f : (0,T) \times \otimes_{k=0}^\kappa \mathbb{R}^{p_k} \to \mathbb{R}$ via $f(t,v)(x) = f(t,v(x))$. In the setup discussed in Section 3, we aim to determine a symbolic expression for $f$ using parameterized networks $\mathfrak{S}_\sigma^\theta$ of the form in (1). Since $\mathfrak{S}_\sigma^\theta$ operates pointwise in time and space, we require the operator $\mathcal{J}_\kappa$ for the parameterizations $\mathfrak{S}_\sigma^\theta$ to depend on derivatives of the state. In the following, we specify the framework outlined above on the basis of (Holler & Morina, 2024a, Subsection 2.1).

**Assumption 4** (Space setup). *Suppose that the spaces $V_k$, for $1 \leq k \leq \kappa$, the state space $V$, the image space $W$ and the space $\tilde{V}$ are reflexive, separable Banach spaces. Assume for some $1 \leq \hat{q} \leq \hat{p} < \infty$ the embeddings*

$$V \hookrightarrow \tilde{V} \hookrightarrow W, \ V \hookrightarrow\!\!\!\rightarrow W^{\kappa,\hat{p}}(\Omega), \ V \hookrightarrow Y, \ L^{\hat{q}}(\Omega) \hookrightarrow W,$$

$$L^{\hat{p}}(\Omega) \hookrightarrow V_k \hookrightarrow L^{\hat{q}}(\Omega), \ \text{for } 1 \leq k \leq \kappa, \ \text{and either } W^{\kappa,\hat{p}}(\Omega) \hookrightarrow \tilde{V} \ \text{or } \tilde{V} \hookrightarrow W^{\kappa,\hat{p}}(\Omega).$$

*The dynamic spaces are defined as Sobolev-Bochner spaces (Roubíček, 2012, Chapter 7), by*

$$\mathcal{V} = L^p(0,T;V) \cap W^{1,p,p}(0,T;\tilde{V}), \ \mathcal{W} = L^q(0,T;W), \ \mathcal{Y} = L^r(0,T;Y),$$

$$\mathcal{V}_0 = \mathcal{V}_0^\times := \mathcal{V}, \ \mathcal{V}_k = L^p(0,T;V_k), \ \mathcal{V}_k^\times = L^p(0,T;V_k^\times) \ for \ 1 \leq k \leq \kappa$$

*for some $1 \leq p,q,r < \infty$ with $p \geq q$. Finally, for some constant $c_\mathcal{V} > 0$, for all $v \in \mathcal{V}$, we assume uniform state space regularity*

$$\|\mathcal{J}_\kappa v\|_{L^\infty((0,T)\times\Omega)} \leq c_\mathcal{V}\|v\|_\mathcal{V}. \tag{3}$$

**Remark 5** (State space regularity)**.** *The required compact embedding $V \hookrightarrow\hookrightarrow W^{\kappa,\hat{p}}(\Omega)$ holds, e.g., for $V$ being a Sobolev space with suitable parameters, such that the Rellich-Kondrachov Theorem (see (Adams & Fournier, 2003, Theorem 6.3) and (Evans, 2010, §5.7)) applies. The extended state space embedding behind* (3) *follows for $\tilde{V}$ being a sufficiently regular Sobolev space by (Roubíček, 2012, Lemma 7.1). We refer to (Holler & Morina, 2024a, Remark 9) for the details.*

### 3.4.2 Regularity

We justify that the parameterized symbolic networks $\mathfrak{S}_\sigma^\theta$, of the form as introduced in (1), can be extended to corresponding operators acting on underlying function spaces. Furthermore, we establish first regularity properties of these operators, formulated in function space. Note that these properties are necessary for (Holler & Morina, 2024a, Assumption 3) to hold true. To this end, we apply analogous arguments as in (Aarset et al., 2023, Lemma 4) for the extension to function space and (Holler & Morina, 2024a, Lemma 17) for the regularity result. The proof of the following statement is provided in Appendix A.

**Proposition 6.** *Let Assumption 4 hold true. Then the symbolic network $\mathfrak{S}_\sigma^\theta : \mathbb{R}^{n_0^\sigma} \to \mathbb{R}$ induces well-defined Nemytskii operators $\mathfrak{S}_\sigma^\theta : \otimes_{k=0}^\kappa \mathcal{V}_k^\times \to L^q(0,T;L^{\hat{q}}(\Omega))$ and $\mathfrak{S}_\sigma^\theta : \otimes_{k=0}^\kappa \mathcal{V}_k^\times \to \mathcal{W}$, both via $[\mathfrak{S}_\sigma^\theta(u)](t) = \mathfrak{S}_\sigma^\theta(u(t,\cdot))$. Furthermore, if the $(\sigma_l)_{1 \leq l \leq L}$ are locally Lipschitz continuous, then*

$$\mathfrak{S} : \Theta \times \mathcal{V} \to \mathcal{W}$$

$$(\theta, v) \mapsto \mathfrak{S}(\theta, v) =: \mathfrak{S}_\sigma^\theta(\mathcal{J}_\kappa v)$$

*is weak-strong continuous. Moreover, for bounded $U \subset \mathbb{R}^{n_0^\sigma}$, the map given by*

$$\Theta \ni \theta \to \mathfrak{S}_\sigma^\theta \in L^\infty(U)$$

*is continuous.*

*Proof.* See Appendix A. $\qquad\qquad\square$

## 4 Symbolic model learning

The central objective of this work is the identification of a (partially) unknown state $u^\dagger \in \mathcal{V}$ and a corresponding physical law $f^\dagger : \otimes_{k=0}^\kappa \mathcal{V}_k^\times \to \mathcal{W}$, ideally expressed in a symbolically simple form, that satisfies the partial differential equation

$$\begin{aligned}
\partial_t u(t,x) &= f(t, \mathcal{J}_\kappa u(t,x)), \quad \text{for } (t,x) \in (0,T) \times \Omega, \\
\text{s.t. } K^\dagger u &= y^\dagger.
\end{aligned} \tag{$\mathcal{E}$}$$

Here, $\mathcal{Y} \ni y^\dagger = K^\dagger u^\dagger$ denotes the full measurement data and $K^\dagger$ is the measurement operator mapping the state $u^\dagger$ to the observation $y^\dagger$. The measurement data $y$ is modeled as $y : (0, T) \to Y$, with a static measurement space $Y$ and time extension $\mathcal{Y}$. In practice, the data $y^\dagger$ is provided via approximate measurements $\mathcal{Y} \ni y^m \approx K^m u^\dagger$, where $K^m : \mathcal{V} \to \mathcal{Y}$, for $m \in \mathbb{N}$, are reduced measurement operators. The measurements are further assumed to satisfy the noise estimate

$$\|y^m - K^m u^\dagger\|_{\mathcal{Y}}^r \leq \delta(m), \tag{4}$$

where $\delta : \mathbb{N} \to \mathbb{R}_{\geq 0}$ fulfills $\lim_{m \to \infty} \delta(m) = 0$.

## 4.1 Framework

For the purpose of reconstruction, we approximate the physical law using symbolic networks $\mathfrak{S}_\sigma^\theta$, parameterized by $\theta \in \Theta$, as introduced in (1) in Section 3. In this work, we adopt the assumption that $f^\dagger \in \left\{ \mathfrak{S}_\sigma^\theta \mid \theta \in \Theta \right\}$ is representable by a fixed architecture and briefly discuss a more general setup in the context of Holler & Morina (2024a) later. Motivated by Holler & Morina (2024a) and following an approach similar to that in Scholl et al. (2025), we aim to solve the reconstruction problem using the all-at-once formulation

$$\min_{u \in \mathcal{V}, \theta \in \Theta} \lambda^m \|\partial_t u - \mathfrak{S}_\sigma^\theta (\mathcal{J}_\kappa u)\|_{\mathcal{W}}^q + \mu^m \|K^m u - y^m\|_{\mathcal{Y}}^r + \mathcal{R}(u, \theta) \tag{$\mathcal{P}^m$}$$

where $\lambda^m, \mu^m > 0$ for $m \in \mathbb{N}$, and $\mathcal{R}$ denotes a suitable regularization functional. We next detail the abstract conditions necessary for the well-posedness of $(\mathcal{P}^m)$ and our main identification results, followed by a discussion of their practical application and relevance.

### 4.1.1 Structural requirements

In addition to Assumption 4, we pose the following assumptions on the data acquisition, the existence of an admissible solution to $(\mathcal{E})$, and the regularization.

**Assumption 7.**

_Measurements._ _Let the full measurement operator $K^\dagger$ be injective and weak-strong continuous. Suppose further that the operators $K^m : \mathcal{V} \to \mathcal{Y}$ are weak-weak continuous, for $m \in \mathbb{N}$, and that for any weakly convergent sequence $(u^m)_m \subset \mathcal{V}$_

$$K^m u^m - K^\dagger u^m \to 0 \quad in \ \mathcal{Y} \quad as \ m \to \infty. \tag{5}$$

_Admissible solution._ _Assume that for the given full measurement data $y^\dagger \in \mathcal{Y}$, there exists $f^\dagger : \otimes_{k=0}^\kappa \mathcal{V}_k^\times \to \mathcal{W}$, to be understood as the space-time extension of an underlying scalar version $f^\dagger : (0, T) \times \otimes_{k=0}^\kappa \mathbb{R}^{p_k} \to \mathbb{R}$, and $u^\dagger \in \mathcal{V}$ fulfilling $(\mathcal{E})$._

_Regularization._ _Suppose that the regularization functional $\mathcal{R}$ is of the form_

$$\mathcal{R} : \mathcal{V} \times \Theta \to [0, \infty], \quad \mathcal{R}(u, \theta) = \|u\|_{\mathcal{V}}^p + \mathcal{R}_0(\theta),$$

_for $\mathcal{R}_0 : \Theta \to [0, \infty]$ a proper, coercive and weakly lower semicontinuous functional._

As a first step toward establishing the reconstructibility of a solution to $(\mathcal{E})$ via solutions to $(\mathcal{P}^m)$, we argue well-posedness of $(\mathcal{P}^m)$ under Assumption 4 and 7.

**Lemma 8.** *Under Assumption 4 and 7 problem* $(\mathcal{P}^m)$ *is well-posed for* $m \in \mathbb{N}$.

*Proof.* See Appendix B for the details. □

Before proceeding to the main reconstruction results, we discuss the practical implications of the requirements in Assumption 7, specifically addressing their alignment with realistic scenarios.

### 4.1.2 Discussion of assumptions

The theoretical results of this work rely on Assumptions 4 and 7. While these assumptions are analytically motivated, it is important to discuss their relevance.

**Regularity.** The functional-analytic setting introduced in Assumption 4 is standard for the considered PDE reconstruction problem. The more restrictive aspects are (i) the compact embedding of the state space $V$ into $W^{\kappa,\hat{p}}(\Omega)$ and (ii) the uniform state space embedding in (3). Regarding (i), recall that $\kappa$ denotes the highest order of derivatives on which the unknown model depends. Essentially, reconstructing a model that depends on derivatives of order $\kappa$ requires the state space to possess a regularity order greater than $\kappa$ on the state space which is also necessary for (3), see (Holler & Morina, 2024a, Remark 9). These assumptions are not only required for the reconstruction results of this work, but also to guarantee the existence of a solution to the minimization problem $(\mathcal{P}^m)$. Since the symbolic network $\mathfrak{S}_\sigma^\theta$ is highly nonlinear, additional regularity in the state space is required to ensure the continuity properties necessary for stability. From a practical standpoint, this is intuitive, as one cannot expect to solve the inverse problem using the same minimal regularity assumptions as for a standard PDE forward problem. However, these requirements also impose stricter conditions on what constitutes an *admissible solution* under Assumption 7. While the existence of a solution to $(\mathcal{E})$ is a logical baseline for the reconstruction itself, the specific demand for high space regularity remains the primary analytical hurdle, and its practical feasibility is discussed in the next paragraph.

**Admissible solution.** Ensuring the existence of an admissible solution to $(\mathcal{E})$ with state regularity $\mathcal{V}$, as required by Assumption 7, can be challenging in practice, given the regularity typically provided by the underlying equation. For example, in the case of the transport equation, this issue is discussed after (Holler & Morina, 2024a, Proposition 1) (see, e.g., Ambrosio (2017); DiPerna & Lions (1989) for well-posedness results). It is important to note, however, that this requirement is implicitly an assumption on the model $f$, since the regularity of the state is generally inherited from the regularity of the model itself. For the transport equation for example, Brué & Nguyen (2020) shows a propagation of Sobolev regularity under regularity assumptions on the drift.

**Measurements.** The measurement framework, designed such that restrictive conditions are shifted to the full measurement operator $K^\dagger$ associated with $(\mathcal{E})$, includes the following conditions.

- *Continuity properties:* We require $K^\dagger$ to satisfy *weak-strong continuity*. This is a stronger condition than the *weak-weak continuity* required for the reduced measurement operators $(K^m)_m$ in $(\mathcal{P}^m)$, which is essential to guarantee the existence of a minimizer for the problem $(\mathcal{P}^m)$. Note that for linear $(K^m)_m$ weak-weak continuity is equivalent to continuity.

- *Injectivity:* By imposing injectivity on $K^\dagger$, we ensure that the state of $(\mathcal{E})$ to be reconstructed is uniquely determined by some $u^\dagger$. In the ensuing reconstruction results, $u^\dagger$ is approximated by the sequence of states $(u^m)_m$ solving $(\mathcal{P}^m)$. Without injectivity, one would be forced to adopt a weaker notion of approximation.

- *Approximation:* It is important to note that the reduced measurement operators $(K^m)_m$ need not be injective nor possess enhanced regularity beyond weak-weak continuity. For instance, one may choose $K^\dagger$ as the embedding operator, modeling ideal full-resolution observations, and $(K^m)_m$ as low-resolution sampling operators, defined as in (30).

  The operators $(K^m)_m$ are related to $K^\dagger$ via the abstract convergence condition (5), which encompasses a broad range of scenarios. It is applicable to sequences of bounded linear operators $(K^m)_m$ that converge to $K^\dagger$ in the operator norm. In case of nonlinear operators, this concept can be extended to sequences $(K^m)_m$ that converge uniformly to $K^\dagger$ on bounded subsets of $\mathcal{V}$. For the sampling operators in (30), the general condition (5) is satisfied (see Appendix C).

- *Noise model:* The only requirement is (4), which is flexible as it imposes no assumptions on the specific distribution or nature of the noise.

**Regularization.** The assumptions on the regularization framework in Assumption 7 are standard in variational regularization (see, e.g., Scherzer et al. (2008)). The functional $\mathcal{R}_0$ allows to incorporate prior information on the parameters $\theta$. A typical choice is the $L^1$-norm, which promotes sparsity in the parameters defining $f_\theta$ Tibshirani (1996). This, in turn, leads to a more concise and interpretable symbolic representation of the reconstructed model (see, e.g., Scholl et al. (2025)). In line with this, the numerical experiments in Section 5 employ the choice $\mathcal{R}_0(\cdot) = \|\cdot\|_{L^1}$.

## 4.2 Main results

In the following, we examine the problem of physical law learning, where an admissible state $u^\dagger$ and physical law $f^\dagger$ are reconstructed by $u^m$ and $\mathfrak{S}_\sigma^{\theta^m}$ for $m \in \mathbb{N}$, respectively, obtained from $(u^m, \theta^m)$ solving $(\mathcal{P}^m)$. The analysis focuses on the setup where $f^\dagger$ can be represented within a predefined architecture parameterized by a set $\Theta$. This approach relies on the assumption that $f^\dagger$ admits a concise and interpretable symbolic representation and presumes that the chosen architecture, designed by the user (e.g., ParFam from Scholl et al. (2025)), is sufficiently expressive to capture such representations. We state our main result:

**Theorem 9.** *Let Assumption 4 and 7 hold true and suppose that there exists*

$$f^\dagger \in \left\{ \mathfrak{S}_\sigma^\theta \mid \theta \in \Theta \right\},$$

*admissible to $(\mathcal{E})$. Let $(u^m, \theta^m)$ solve $(\mathcal{P}^m)$ for $m \in \mathbb{N}$ and $\lambda^m, \mu^m > 0$ such that*

$$\lambda^m \to \infty \quad and \quad \mu^m \to \infty \quad with \quad \mu^m \delta(m) \to 0 \quad as \quad m \to \infty.$$

*Then there exists a subsequence $(\theta^{m_l})_l$ converging to some $\tilde{\theta} \in \Theta$ such that*

$$\partial_t u^\dagger = \mathfrak{S}_\sigma^{\tilde{\theta}}(\mathcal{J}_\kappa u^\dagger) \quad and \quad \mathfrak{S}_\sigma^{\theta^{m_l}} \to \mathfrak{S}_\sigma^{\tilde{\theta}} \ in \ L_{loc}^\infty(\mathbb{R}^{n_0^\sigma}) \ as \ l \to \infty. \tag{6}$$

*This holds for any convergent subsequence. In addition also $u^m \rightharpoonup u^\dagger$ as $m \to \infty$.*

*Moreover, the tuple $(u^\dagger, \tilde{\theta})$ is a regularization minimizing solution, i.e., for all $(u^\dagger, \theta^\dagger)$ solving $\partial_t u^\dagger = \mathfrak{S}_\sigma^{\theta^\dagger}(\mathcal{J}_\kappa u^\dagger)$ it holds true that $\mathcal{R}(u^\dagger, \tilde{\theta}) \leq \mathcal{R}(u^\dagger, \theta^\dagger)$.*

*Proof.* See Appendix B for the details. □

The concluding assertion of Theorem 9 is particularly interesting from a practical standpoint. Regularization of the parameter $\theta$ in the $L^1$-norm promotes the reconstruction of a sparse parameterization of $f^\dagger$ through the solution of $(\mathcal{P}^m)$. This parameterization can be understood as a concise symbolic expression of the underlying physical law, offering enhanced interpretability. Another important observation is that if an appropriate identifiability condition, as established in Scholl et al. (2023a), holds, a stronger result can be obtained. Specifically, Scholl et al. (2023a) demonstrates that the unique identifiability of a linear/algebraic function $f$ based on full state measurements is equivalent to the linear/algebraic independence of the state variables (e.g., derivatives up to order $\kappa$) on which $f$ acts. Here, we state the *identifiability condition* under consideration in a general formulation as follows:

$$\text{For} \quad f_1, f_2 : \otimes_{k=0}^{\kappa} \mathcal{V}_k^\times \to \mathcal{W} \quad \text{with} \quad f_1(\mathcal{J}_\kappa u^\dagger) = f_2(\mathcal{J}_\kappa u^\dagger) \quad \Rightarrow \quad f_1 = f_2 \qquad (\mathcal{I})$$

Assuming condition $(\mathcal{I})$, the convergence in (6) holds for the entire sequence $(\theta^m)_m$.

**Corollary 10.** *Let the assumptions of Theorem 9 and $(\mathcal{I})$ apply. Then*

$$\mathfrak{S}_\sigma^{\theta^m} \to \mathfrak{S}_\sigma^{\theta^\dagger} = f^\dagger \quad in \ L_{loc}^\infty(\mathbb{R}^{n_0^\sigma}) \ as \ m \to \infty.$$

*Proof.* See Appendix B for the details. □

The result in Theorem 9 shows that the symbolic networks $(\mathfrak{S}_\sigma^{\theta^m})_m$ associated with the minimization problems $(\mathcal{P}^m)$ recover physical laws that are consistent with the PDE $(\mathcal{E})$ for the state $u^\dagger$, in the sense of subsequential convergence. More precisely, for every convergent subsequence of $(\theta^m)_m$ (and there exists one) the corresponding sequence of symbolic networks converges to a physical law consistent with the PDE. While we cannot guarantee the entire sequence of symbolic networks will converge when multiple physical laws are valid (since two subsequences could recover two different admissible physical laws), our results in fact establish a stronger claim than only subsequential convergence. The reconstructed physical laws are limited to those whose parameters, in their symbolic network representation, minimize the regularization functional. When a sparsity-promoting regularizer is used, the system preferentially recovers *simple* physical laws characterized by concise symbolic representations. If the ground truth law is unique, e.g., under an identifiability condition such as $(\mathcal{I})$, then the entire sequence $(\mathfrak{S}_\sigma^{\theta^m})_m$ converges to the unique physical law. Notably, the convergence statements hold uniformly on compact sets rather than merely pointwise.

### 4.3 Extension of results

The results of Theorem 9 and Corollary 10 are derived under the rigid architectural assumption that $f^\dagger \in \{\mathfrak{S}_\sigma^\theta \mid \theta \in \Theta\}$. This is the same regime considered in Lampert & Martius (2017), where the EQL architecture for neural-network-based symbolic regression is introduced. In that setting, non-representability of the underlying physical law, illustrated for the cart-pendulum system in (Lampert & Martius, 2017, System (13)), leads to poor extrapolation performance unless the architecture is enlarged, as demonstrated in (Sahoo et al., 2018, Subsection 4.4).

The main results presented here in Subsection 4.2 extend directly beyond this rigid architectural assumption to more general settings. One such generalization considers the case where, instead of

a fixed architecture $\mathfrak{S}_\sigma^\theta$ for $\theta \in \Theta$, we consider growing architectures $\mathfrak{S}_{\sigma^m}^{\theta,m}$ for $\theta \in \Theta^m$ such that $f^\dagger$ is only representable by some $\mathfrak{S}_{\sigma^m}^{\theta,m}$ for sufficiently large $m \in \mathbb{N}$.

Another scenario is the case where admissible physical laws $f^\dagger$ are not representable by any finite-dimensional architecture, regardless of its size. We provide a brief roadmap for extending our results to this setting. This occurs when $f^\dagger$ is not expressible in a symbolically concise form (e.g., the Gaussian error function) or when the architecture is not designed in a suitable way, e.g., a purely rational network. Such networks essentially represent rational functions and inherently fail to express base functions such as the exponential. From a practical perspective, along with theoretical results on the approximation properties of symbolic networks Holler & Morina (2025a), it is natural to approximate $f^\dagger$ using growing architectures in the limit as regularization-minimizing solution. This approach aligns with Holler & Morina (2024a), which study unique reconstructibility in this generalized setting (see also Aarset et al. (2023); Kaltenbacher (2016); Kaltenbacher & Nguyen (2022) for related all-at-once formulations). However, two technical challenges associated with these reconstructions require careful consideration (see (Holler & Morina, 2024a, Assumption 2-5)), namely i) the weak lower semicontinuity of the $\mathcal{C}^1(U)$-seminorm of $\mathfrak{S}_\sigma^\theta$ for bounded $U \subseteq \mathbb{R}^{n_0^\sigma}$ with respect to the parameterization $\theta \in \Theta$ and ii) an approximation capacity condition as in (Holler & Morina, 2024a, Assumption 5 iii)). We believe that i) can be similarly obtained following the arguments proving Proposition 6 under additional notational technicalities. For ii), the underlying approximation (rate) follows for symbolic networks from the considerations in Holler & Morina (2025a) on first order approximation results in case the target function is sufficiently regular. Although the growth of the parameters that realize the approximating networks, as considered in the work Holler & Morina (2024b), is not addressed there, we believe that this can also be achieved with technical effort. Once a regularization-minimizing solution $\tilde{f}$ has been reconstructed, a practical strategy to refine its representation involves identifying a suitable parameterization $f_{\tilde{\theta}} \approx \tilde{f}$. This can be accomplished using a fixed, expressive symbolic network and applying $L^1$-regularization to $\tilde{\theta}$, promoting sparsity in the symbolic representation.

## 5 Numerical experiments

In this section, we present a numerical setup to illustrate the analytical identification results established in Section 4. Specifically, we aim to evaluate whether the results predicted by Theorem 9 and Corollary 10 can be observed in practice. To achieve this, we consider three settings:

1) A **simplified analytical setting** focusing on the identification of unknown physical laws $f$ and states $u$ using one-dimensional linear PDEs of the form

$$\partial_t u = f(t, u, \partial_x u),$$

where $f$ depends on the state $u$ and its first spatial derivative $\partial_x u$. This simplified framework allows us to test the validity and practical applicability of the theoretical results within a controlled computational environment.

2) The **Strogatz datasets** (see, e.g., (La Cava et al., 2021, Table 3)), a standard test suite for symbolic regression in dynamical systems, are used to evaluate our approach. This suite comprises seven physically motivated, first-order nonlinear ODE systems that exhibit chaotic behavior and span a range of complexities ideal for symbolic model learning.

3) Two different **two-dimensional PDE systems** are considered to further evaluate our approach.

While this work focuses primarily on the theoretical reconstructibility results in function space (Section 4), our numerical experiments are intended to demonstrate the framework's practical utility. By evaluating the Strogatz datasets and two distinct 2D PDE systems, we move beyond simple one-dimensional linear cases to test the method against more sophisticated regimes. This approach not only supports the relevance of the underlying theory but also helps identify potential boundary cases and limitations. All numerical experiments of this work can be reproduced based on the publicly available source code[2].

### 5.1 Implementation framework

We now provide the analytical and numerical framework for the three settings as describe above. For the setting 1), the *simplified analytical setting*, we provide the functional-analytic background in view of our analytic results in the next subsection. After that, Subsection 5.1.2 provides the numerical setup used for all experiments of the settings 1)-3).

#### 5.1.1 Analysis setup

The analytic setup discussed next, is being configured for the one-dimensional linear PDEs considered for setting 1). In view of Assumption 4, we adopt a Hilbert space framework with parameters $p = q = \hat{p} = \hat{q} = r = 2$, domain $\Omega = [0,4]$, time $T = 3$, state space $V = H^2(\Omega)$, image space $W = L^2(\Omega)$, measurement space $Y = L^2(\Omega)$ and the corresponding dynamic spaces as defined in Assumption 4. The state space $V$ is assumed to have higher regularity to ensure a compact embedding into $H^1(\Omega)$, which is crucial, since the unknown physical law depends on up to $\kappa = 1$ spatial derivatives of state $u$ (compare with Subsection 4.1). Similarly, the space $\tilde{V}$ requires additional regularity to guarantee condition (3), as ensured by (Holler & Morina, 2024a, Remark 9) for $\tilde{V} = H^2(\Omega)$. We further choose the full measurement operator $K^\dagger$ as the embedding operator $\iota : \mathcal{V} \to \mathcal{Y}$, i.e., $K^\dagger u = \iota u$. This operator is injective and weak-strong continuous, due to the compact embedding $\mathcal{V} \hookrightarrow \mathcal{Y}$, as guaranteed by the Aubin-Lions Lemma (Roubíček, 2012, Lemma 7.7). Specifically, with $\langle \cdot, \cdot \rangle$ denoting the inner product in $L^2(\Omega)$ and an orthonormal system $(e_i)_{i \in \mathbb{N}}$ of $L^2(\Omega)$ (e.g., for $\Omega = [0,1]$, $\Omega \ni x \mapsto e_i(x) = \cos(i\pi x) \in L^2(\Omega)$), we can express $K^\dagger$ by

$$K^\dagger : \mathcal{V} \to \mathcal{Y}, \quad [K^\dagger u](t) = \sum_{i \in \mathbb{N}} \langle e_i, u(t) \rangle e_i$$

for $u \in \mathcal{V}$ and $t \in [0,T]$. The reduced measurement operators, corresponding to low-frequency sampling operators (or truncated Fourier measurement operators), are given as follows. Let $0 = t_1 < t_2 < \cdots < t_m < t_{m+1} = T$ be the $m$-equidistant grid on $[0,T]$ for $m \in [0,T]$ and $\Delta_m := T/m$. Then, for $1 \le j \le m-1$ and $t \in [t_j, t_{j+1})$ or $j = m$ and $t \in [t_m, t_{m+1}]$, we define $K^m$ for $m \in \mathbb{N}$ by

$$K^m : \mathcal{V} \to \mathcal{Y}, \quad [K^m u](t) = \sum_{i=1}^{m} \left( \Delta_m^{-1} \int_{t_j}^{t_{j+1}} \langle e_i, u(s) \rangle \, \mathrm{d}s \right) e_i \tag{7}$$

for $u \in \mathcal{V}$. Note that the operator $K^m$ performs time averaging over the respective time intervals. The well-definedness of $K^\dagger$ and $K^m$ for $m \in \mathbb{N}$, along with the convergence condition (5) in Assumption 7, are discussed in detail in Appendix C.

---

[2]See `https://github.com/Philipp238/unique_sym`.

### 5.1.2 Simulation setup

We now detail the concrete choices made for the quantities appearing in Theorem 9, and provide the numerical setup used for all experiments of the settings 1)-3). Since letting $m \to \infty$ is infeasible in practice, we fix a even maximal value $M \in \mathbb{N}$ and consider measurements for $m = M/2, \ldots, M-1$. Guided by the preceding analysis, we apply a low-pass filtering strategy by defining $K^m$ to retain the lowest $m/M$ fraction of frequency components. Throughout all experiments we set $q = r = 2$ and use the parameter choices

$$\lambda^m = \frac{m^3}{M}, \qquad \mu^m = \frac{m}{M}, \qquad \delta^m = 0.1 \frac{M}{m^3}. \tag{8}$$

Measurement noise is simulated by multiplicative noise according to

$$y^m = K^m(u^\dagger) \cdot \epsilon^{\delta^m}, \tag{9}$$

with uniform perturbations $\epsilon \sim \mathrm{Unif}(0.5, 1.5)$. As $\delta^m$ decreases with $m$, the factor $\epsilon^{\delta^m}$ approaches 1 for large $m$, producing weaker distortion. We employ multiplicative noise to reflect the wide dynamic range of the solution function, ensuring that small values are perturbed less strongly than large ones.

Following Theorem 9, we model $f_\theta$ using ParFam Scholl et al. (2025) and represent $u$ as function evaluations on a discretized grid. The ParFam architecture, introduced in Equation (1), adopts the practical design of Scholl et al. (2025), who observed that a single hidden layer ($L = 1$) is sufficient for expressive modeling while simplifying optimization. We select numerator polynomials of degree 3, denominator polynomials of degree 2, and sine and exponential activation functions. ParFam permits custom loss functions, which we specify using the objective of ($\mathcal{P}^m$). Parameter optimization is performed using basin-hopping Wales & Doye (1997) combined with BFGS, both run for 100 iterations with default settings otherwise.

To approximately solve the minimization problem ($\mathcal{P}^m$), we employ an alternating optimization scheme. Specifically, we iteratively optimize with respect to $\theta$, keeping $u$ fixed, and then optimize with respect to $u$ while fixing $\theta$. The parameter $\theta$ is updated using the ParFam procedure described above, while $u$ is trained using ADAM Kingma & Ba (2015) with a learning rate of 0.0004, up to 300 iterations per step (linear 1D PDE and ODE experiments) or 100 iterations (2D PDE experiments), with early stopping (patience 10). This alternating optimization scheme is repeated for 100 full cycles. Time derivatives are estimated using central finite differences (linear 1D PDE experiments) or Savitzky-Golay filtering (ODE and 2D PDE experiments) for robustness to noise. Table 1 summarizes the shared optimization hyperparameters, while the ParFam architecture choices for each experiment are detailed in the respective subsections.

## 5.2 Numerical results

In this subsection, we present a series of numerical results. We start with the setting 1) above, serving as a proof-of-concept for the results discussed in Section 4. Then we present the numerical considerations for the settings 2) and 3), based on more sophisticated differential equations to support the practical relevance of the discussed framework.

| Parameter | Value |
|---|---|
| Alternating optimization epochs | 100 |
| Early stopping patience | 10 |
| State iterations per epoch (1D / 2D) | 300 / 100 |
| Learning rate (ADAM, state $u$) | $4 \times 10^{-4}$ |
| ParFam regularization | $10^{-3}$ |
| Noise model | multiplicative, Eq. (9) |
| $\delta^m$ | $0.1 \, M/m^3$ |
| Maximum $M$ | 100 |
| Seeds per experiment | 10-15 |
| Derivative estimation (1D PDE / ODE, 2D PDE) | finite diff. / Savitzky-Golay |

Table 1: Shared optimization hyperparameters across all experiments.

### 5.2.1 Simplified analytical setting

The following results cover two different one-dimensional linear PDEs, selected from the numerical section in (Scholl et al., 2023a, Subsection 5.1).

**Uniquely identifiable PDE.** First, we focus on an uniquely identifiable PDE, specifically

$$u_t = au + bu_x,$$

which attains the analytical solution $u^\dagger(t, x) = (x + bt) \exp(at)$. We set $a = 1$ and $b = 2$, and sample $u$ on an equidistant grid over the domain $[0, 3] \times [0, 4]$, with 100 grid points along spatial $x$-direction and 80 along temporal $t$-direction. In Figure 2 (a), we report the results for $M = 100$ and $m = 50, \ldots, 99$. To assess the statements from Theorem 9, we present the $L^2$-distance between $f_{\theta^m}(u^\dagger, u_x^\dagger)$ and $u_t^\dagger$, as well as between the learned solution $u^m$ and the ground-truth $u^\dagger$. These results clearly demonstrate the convergence predicted by Theorem 9. Furthermore, since the PDE is uniquely identifiable, Corollary 10 implies that $f_{\theta^m}$ converges to $f^\dagger$ where $f^\dagger(v, w) = v + 2w$. This is supported by the learned approximation:

$$f_{\theta^m}(u, u_x) = 1.006u - 0.005u_x^2 + 2.116u_x - 0.535,$$

for $m = 100$, which closely matches the true operator. Note that in Figure 2 (a) the deviation of $f_{\theta^m}$ from $f^\dagger$ is considered on the domain $U_0$ depicted in Figure 2 (c) (the range of $(u^\dagger, u_x^\dagger)$ corresponds to $U$). In Figure 2 (c) this deviation is shown for different choices of the underlying domain.

**Not uniquely identifiable PDE.** As a second example, we consider $u(t, x) = \exp(x - at)$, which solves for any choice of coefficients $a_1, a_2 \in \mathbb{R}$, satisfying $a_1 + a_2 = -a$, the equation

$$u_t = a_1 u + a_2 u_x.$$

We set $a = 1$ and sample $u$ on an equidistant grid over the domain $[0, 3] \times [0, 4]$, with 100 grid points along $x$-direction and 80 grid points along $t$-direction. In Figure 2 (b), we report the results for $M = 100$ and $m = 50, \ldots, 99$. To assess the statements from Theorem 9, we present the $L^2$-distance between $f_{\theta^m}(u^\dagger, u_x^\dagger)$ and $u_t^\dagger$, as well as between the learned solution $u^m$ and the ground-truth $u^\dagger$. These results clearly demonstrate the convergence predicted by Theorem 9.

Since $u^\dagger$ does not solve a unique PDE, we cannot employ Corollary 10. Nevertheless, the result in Theorem 9 guarantees that the learned parameters $\theta^m$ converge to a $L^1$-minimizing parameterization of the underlying PDE model $f_{\theta^m}$ which is solved by $u^\dagger$. This is supported by the learned approximation for $m = 100$:

$$f_{\theta^m}(u, u_x) = -0.982u - 0.016u_x.$$

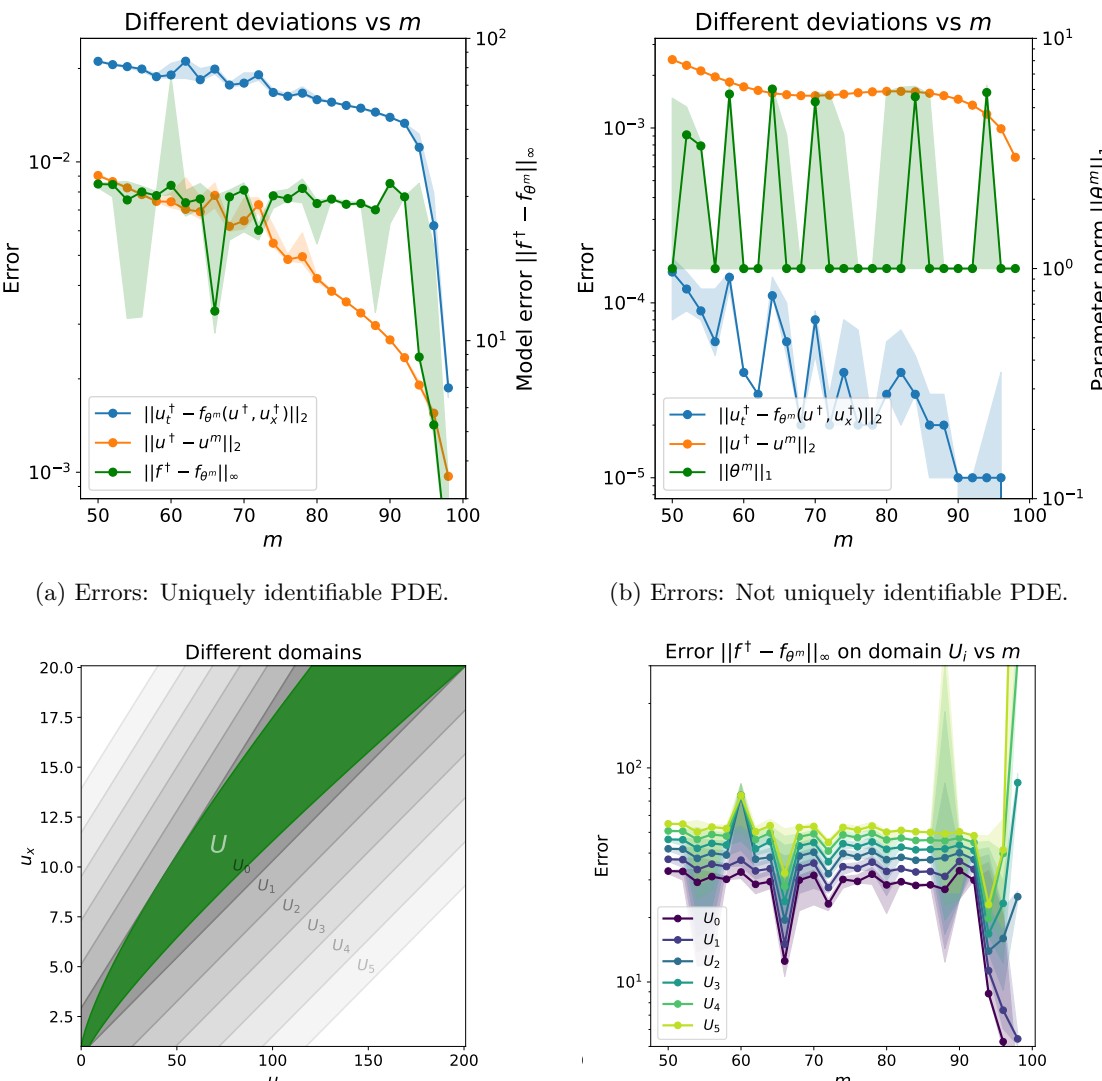

(a) Errors: Uniquely identifiable PDE.

(b) Errors: Not uniquely identifiable PDE.

(c) Model error for different domains for the uniquely identifiable PDE.

Figure 2: Numerical performance. In (a)-(c) the median performance over 10 random seeds with shaded interquartile ranges (IQR) is reported.

### 5.2.2 Strogatz datasets

To evaluate the method on a diverse set of nonlinear dynamics beyond one-dimensional linear PDEs selected from (Scholl et al., 2023a, Subsection 5.1), we consider seven ODE systems from the Strogatz datasets (La Cava et al., 2021, Table 3). These systems, summarized in Table 2, cover polynomial, rational, and trigonometric nonlinearities with varying degrees of complexity.

| System | $f_x$ | $f_y$ |
|---|---|---|
| Bacterial Respiration | $-\frac{xy}{0.5x^2+1} - x + 20$ | $-\frac{xy}{0.5x^2+1} + 10$ |
| Bar Magnets | $-\sin(x) + 0.5\sin(x-y)$ | $-\sin(y) - 0.5\sin(x-y)$ |
| Glider | $-0.05\,x^2 - \sin(y)$ | $x - \cos(y)/x$ |
| Lotka-Volterra | $-x^2 - 2xy + 3x$ | $-xy - y^2 + 2y$ |
| Predator-Prey | $x\left(-x - \frac{y}{x+1} + 4\right)$ | $y\left(\frac{x}{x+1} - 0.075\,y\right)$ |
| Shear Flow | $\frac{\cos(x)\cos(y)}{\sin(y)}$ | $(0.1\sin^2(y) + \cos^2(y))\sin(x)$ |
| Van der Pol | $-\frac{10}{3}x^3 + \frac{10}{3}x + 10y$ | $-x/10$ |

Table 2: Strogatz ODE systems. All systems have the form $\dot{x} = f_x(x,y)$, $\dot{y} = f_y(x,y)$.

For each system, we generate 10-80 trajectories from random initial conditions (depending on the system complexity; see Table 3) and apply the measurement operator $K^m$ in (7) for $m = 50, 52, \ldots, 100$ (26 values). Each $(m, \text{system})$-combination is run for 15 random seeds.

The ParFam architecture is tailored to each system based on the functional form. Polynomial systems use purely polynomial architectures, while trigonometric systems include sin and cos base functions, and rational systems use nonzero denominator degrees. Table 3 summarizes the architecture choices. Note that in all experiments, we apply a lightweight post-processing step, i.e., coefficients with magnitude below 0.015 are set to zero in the recovered formula. This primarily benefits experiments with rational expressions (ODE systems with denominators), where spurious small terms in denominators can cause large extrapolation errors when evaluating $\|f^\dagger - f_{\theta^m}\|_\infty$. For purely polynomial formulas, the effect is minor. This post-processing affects only the error evaluation, not the optimization itself. The post-processing is also applied for the two-dimensional PDEs discussed later.

The results for all seven systems are shown in Figure 3. Note that the state corresponds to $u(t) = (x(t), y(t))$ for $t \geq 0$. For each system, the left subplot displays the PDE residual $\|u_t^\dagger - f_{\theta^m}(u^\dagger)\|_2$, the state error $\|u^\dagger - u^m\|_2$, and the model error $\|f^\dagger - f_{\theta^m}\|_\infty$ (evaluated on a uniform grid covering the bounding box of the trajectory values with a 10% margin) as functions of $m$. The right subplot shows the model error evaluated on $\varepsilon$-neighborhoods around the ground-truth trajectories for various neighborhood widths $\varepsilon$, providing a domain-dependent assessment of formula quality. Across all systems, we observe a decrease in errors as $m$ increases, which is more or less pronounced depending on the system. The polynomial systems (Lotka-Volterra, Van der Pol) are recovered with high accuracy, while the trigonometric and rational systems (Shear Flow, Bacterial Respiration) exhibit larger variability, consistent with their greater functional complexity. In Table 4 we report the best recovered formula among the seeds for $m = 100$.

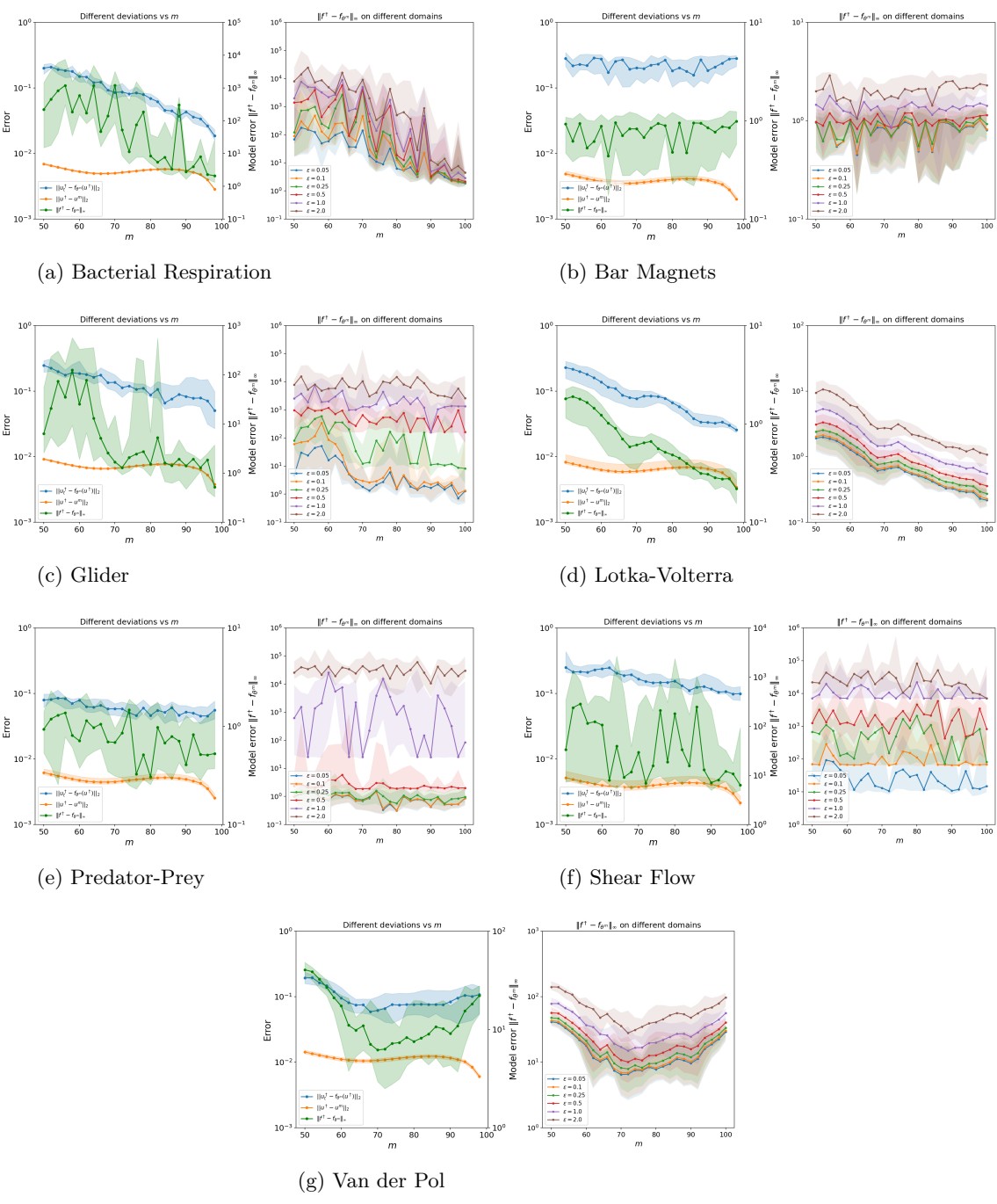

Figure 3: Numerical results for Strogatz ODE systems. For each system (a)-(g), *Left:* PDE residual (blue), state error (orange), and model error (green) vs. $m$ (median $\pm$ IQR over 15 seeds). The model error (left) is evaluated on a uniform grid spanning the range of the training data. *Right:* Model error $\|f^\dagger - f_{\theta^m}\|_\infty$ evaluated on $\varepsilon$-neighborhoods around ground-truth trajectories.

| System | ICs | Num. deg. | Den. deg. | Base fn. | Reps. | Time (s) |
|---|---|---|---|---|---|---|
| Bacterial Resp. $(x/y)$ | 80 | 3 / 2 | 2 / 2 | — | 50 | 1200 |
| Bar Magnets | 50 | 2 | 0 | $\sin, \cos$ | 8 | 45 |
| Glider $(x/y)$ | 50 | 2 / 2 | 0 / 1 | $\sin$ / $\cos$ | 8 | 45 |
| Lotka-Volterra | 10 | 2 | 0 | — | 3 | 20 |
| Predator-Prey | 10 | 3 | 1 | — | 3 | 20 |
| Shear Flow $(x/y)$ | 80 | 2 / 3 | 1 / 0 | $\sin, \cos$ | 8 | 45 |
| Van der Pol | 10 | 3 | 0 | — | 3 | 20 |

Table 3: ParFam architecture per ODE system. **Num. deg.** and **Den. deg.** refer to the output polynomial degrees of the numerator and denominator, respectively. **Base fn.** lists the activation functions used. **ICs** denotes the number of initial conditions. **Reps.** is the number of independent basin-hopping restarts and **Time** is the wall-clock time limit per ParFam fit.

| System | Recovered $\hat{f}$ | $\|f^\dagger - \hat{f}\|_\infty$ |
|---|---|---|
| Bact. Resp. | $\hat{f}_x = (-5.70x^2 - 6.31xy + 0.47x^2 y + \ldots)/(-0.45x^2 - 0.67y^2 - 0.58y + \ldots)$ | 1.95 |
| | $\hat{f}_y = (3.52x^2 + 1.36xy - 0.40y^2 + \ldots)/(0.35x^2 + 0.21xy - 0.16y + 0.90)$ | 0.13 |
| Bar Mag. | $\hat{f}_x = -0.994\sin(x) - 0.495\sin(-0.98x + 1.02y)$ | 0.05 |
| | $\hat{f}_y = 0.548\sin(-0.89x + 1.03y) + 0.958\sin(y)$ | 0.10 |
| Glider | $\hat{f}_x = -0.05x^2 - 0.999\sin(y)$ | 0.01 |
| | $\hat{f}_y = (-x^2 + 1.04\cos(y))/x$ | 0.15 |
| Lotka-Volt. | $\hat{f}_x = -0.97x^2 - 1.97xy + 2.91x$ | 0.14 |
| | $\hat{f}_y = -0.99xy - 0.94y^2 + 1.84y$ | 0.11 |
| Pred.-Prey | $\hat{f}_x = (0.54x^3 - 1.45x^2 + 0.61xy - 2.77x + \ldots)/(-0.48x - 0.88 + \ldots)$ | 0.40 |
| | $\hat{f}_y = (-0.70xy + 0.05xy^2 + 0.05y^2)/(-0.71x - 0.70)$ | 0.03 |
| Shear Flow | $\hat{f}_x = (0.27y\sin(\cdot) + 0.03x\cos(\cdot) + \ldots)/(-0.98\sin(\cdot) - 0.15x + \ldots)$ | 0.89 |
| | $\hat{f}_y = 0.31x - 0.27y\sin(\cdot)\sin(\cdot) + \ldots$ | 1.24 |
| Van der Pol | $\hat{f}_x = -3.36x^3 + 3.36x + 10.02y$ | 1.50 |
| | $\hat{f}_y = -0.1x$ | 0.00 |

Table 4: Recovered formulas at $m=100$ (best seed).

### 5.2.3 Two-dimensional PDE systems

Finally, we discuss several experiments related to setting 3) above on two-dimensional PDE systems.

**Brusselator system.** We now turn to spatially extended systems on the two-dimensional domain $\Omega = [0, L]^2$ with periodic boundary conditions, where the unknown physical law involves the Laplacian of the state. We consider the Brusselator reaction-diffusion system:

$$\partial_t u = D_u \Delta u + a - (b+1)u + u^2 v,$$
$$\partial_t v = D_v \Delta v + bu - u^2 v,$$

where the reaction kinetics $r_u(u,v) = a - (b+1)u + u^2v$ and $r_v(u,v) = bu - u^2v$ are polynomial, and $D_u$, $D_v > 0$ are diffusion coefficients. The forward problem is solved using a Strang splitting scheme: Spectral diffusion (exponential integrator in Fourier space) combined with an 4th-order Runge-Kutta integrator for $r_u$ and $r_v$, on a $64 \times 64$ spatial grid with $L = 50$, $\Delta t = 0.01$, and 80 saved snapshots from 8 initial conditions. The $K^m$ operator corresponds to (7) in two dimensions.

For the different experiments below, we will use up to three difficulty regimes corresponding to increasing nonlinearity, summarized in Table 5. We start with the setting of *unconstrained recovery* as follows, which is evaluated for each of the three difficulty regimes.

| **Regime** | $a$ | $b$ | $D_u$ | $D_v$ |
|---|---|---|---|---|
| Easy | 1 | 3 | 1 | 4 |
| Medium | 2 | 5 | 1 | 5 |
| Hard | 3 | 8 | 1 | 8 |

Table 5: Brusselator parameter regimes. As $b/a$ and $D_v/D_u$ increase, the Turing patterns of the underlying system become more pronounced and the symbolic recovery task becomes harder due to steeper gradients, sharper spatial features and stronger nonlinear reaction-diffusion coupling.

*Unconstrained recovery.* We first consider the recovery of the full right-hand side, including the diffusion terms, without imposing structural constraints. ParFam receives the features $(\Delta u, u, v)$ for the $u$-equation and $(\Delta v, u, v)$ for the $v$-equation, and fits a degree-3 polynomial (maximal potence 2, no denominator, 8 repetitions, 45s time limit per fit). The diffusion coefficients $D_u$, $D_v$ are thus implicitly recovered as the learned coefficients of $\Delta u$ and $\Delta v$. We sweep $m \in \{50, 60, 70, 80, 85, 88, 91, 94, 96, 98, 99, 100\}$ (12 values) with 10 seeds each.

Figure 4 shows the results for all three difficulty regimes summarized in Table 5. For each regime, the $u$- and $v$-equations are shown in separate rows, with the standard error plot on the left and the neighborhood-based model error on the right. The reconstruction is difficult for all regimes, in particular for the $v$-equation of the Brusselator which has a larger diffusion coefficient.

*Effect of architecture choice.* To assess the sensitivity to the ParFam architecture, we compare three configurations on the hard regime: A smaller polynomial architecture (degree 2), the default (degree 3), and an extended architecture including sin base functions and a rational denominator. The results are shown in Figure 5.

The small architecture is too restrictive: It cannot represent the $u^2v$ term and yields flat, high model errors. The extended architecture, while more expressive, only succeeds for high $m$ values ($m \geq 88$) and with fewer seeds, likely due to the enlarged search space. The default architecture (see Figure 4 (c)) provides the best trade-off between expressiveness and optimization reliability.

*Constrained additive structure.* In the previous experiment, ParFam can freely mix the Laplacian with the state variables, potentially producing cross-terms such as $u \cdot \Delta u$. We now enforce the physically motivated additive structure $\partial_t u = D_u \Delta u + r_u(u,v)$ by constraining the Laplacian feature as additive in ParFam (i.e., no cross-terms between $\Delta u$ and $u, v$). The estimated diffusion coefficient is then read off as the learned coefficient of the Laplacian term. All other settings (grid, $m$ values, seeds, ParFam architecture) are the same as for the unconstrained recovery. This corresponds to the assumption that an approximate physical model is available (in this case knowledge of additive

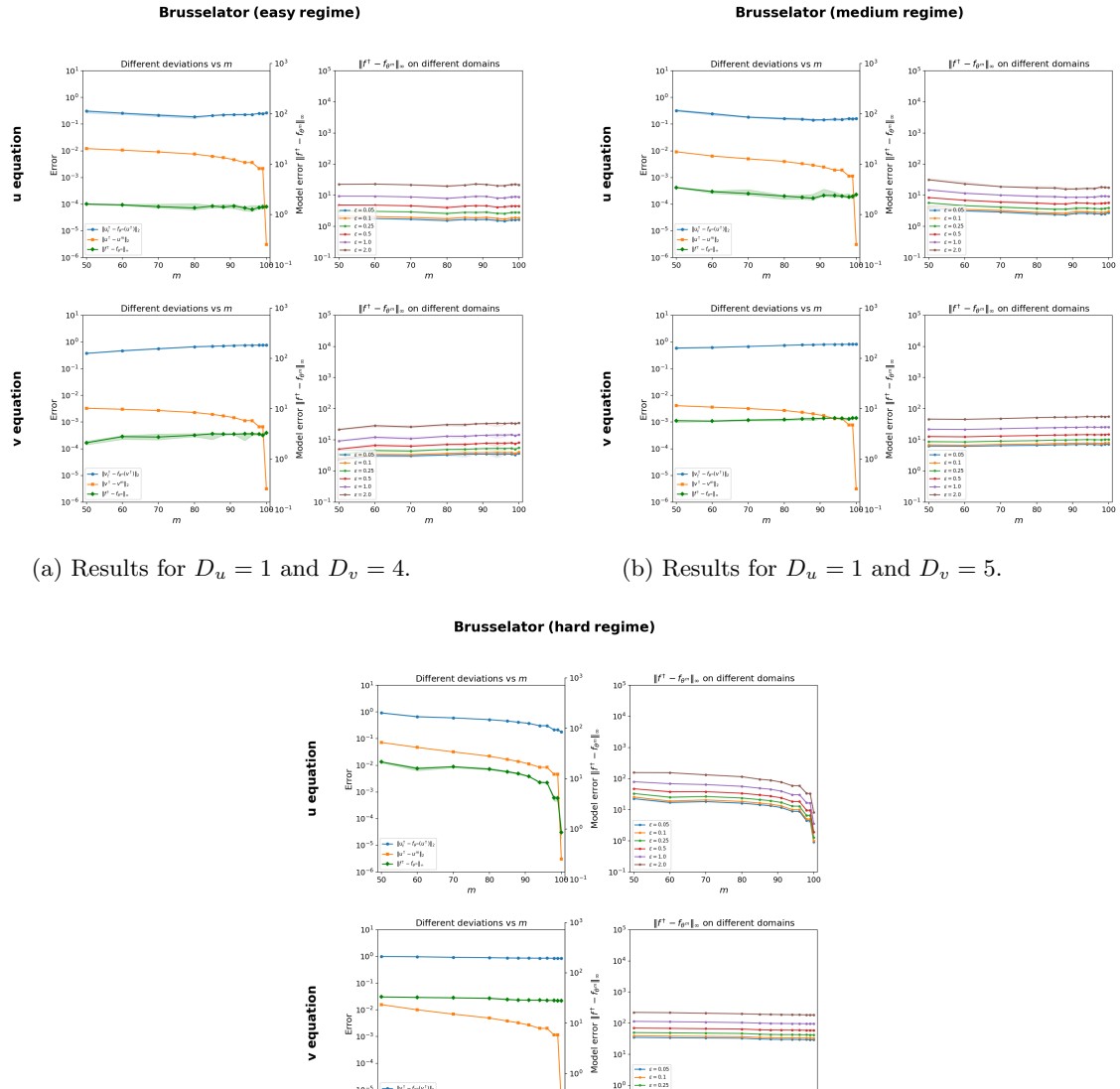

(a) Results for $D_u = 1$ and $D_v = 4$.  (b) Results for $D_u = 1$ and $D_v = 5$.

(c) Results for $D_u = 1$ and $D_v = 8$.

Figure 4: Numerical results for Brusselator. Each subplots shows the $u$-equation (top row) and $v$-equation (bottom row). Left: Error metrics vs. $m$. Right: Model error on $\varepsilon$-neighborhoods.

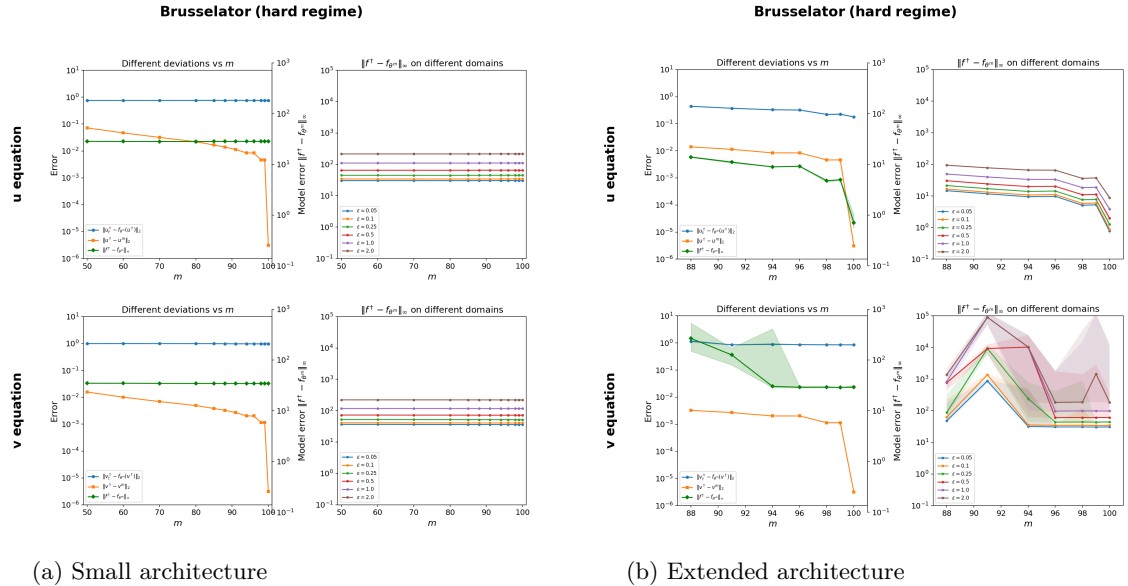

(a) Small architecture          (b) Extended architecture

Figure 5: Architecture comparison on the Brusselator regime with $(D_u = 1, D_v = 8)$. Compare with the default architecture in Figure 4 (c).

diffusion effects) and only *fine-scale* hidden physics (the unknown structure of the reaction kinetics) together with the unknown diffusion coefficients are learned from data.

Figure 6 shows the results for the three different regimes (easy, medium, hard) summarized in Table 5. In addition to the error considered so far, the middle column of subplots displays the convergence of the estimated diffusion coefficients $\hat{D}_u$ and $\hat{D}_v$ toward their true values. The diffusion coefficients are accurately recovered for large $m$ for the easy and the middle regime, whereas for the hard regime in particular the convergence of the $D_v$ coefficient fails. The results indicate that reaction kinetics approximate the correct polynomial form, more or less pronounced depending on the regime.

**Membrane transport.** To evaluate the framework on a qualitatively different PDE system, we consider a model with rational and exponential nonlinearities:

$$\partial_t u = 0.1 \, \Delta u + \frac{v}{1+u} - 0.5 \, u,$$
$$\partial_t v = 0.2 \, \Delta v + e^{-1} - e^{-u} \, v,$$

with equilibrium $(u^*, v^*) = (1, 1)$ on the domain $[0, 10]^2$ with periodic boundary conditions. This system features two distinct nonlinearity types, a rational reaction term $v/(1+u)$ in the $u$-equation and an exponential term $e^{-u}v$ in the $v$-equation, requiring larger ParFam.

The forward problem is solved on a $32 \times 32$ grid using the same Strang splitting scheme as for the Brusselator. We use 8 initial conditions generated from perturbed equilibria with random Gaussian bumps, and sweep $m \in \{50, 60, 70, 80, 85, 88, 91, 94, 96, 98, 100\}$ (11 values) with 10 seeds. The ParFam architectures reflect the structure of each equation, as summarized in Table 6.

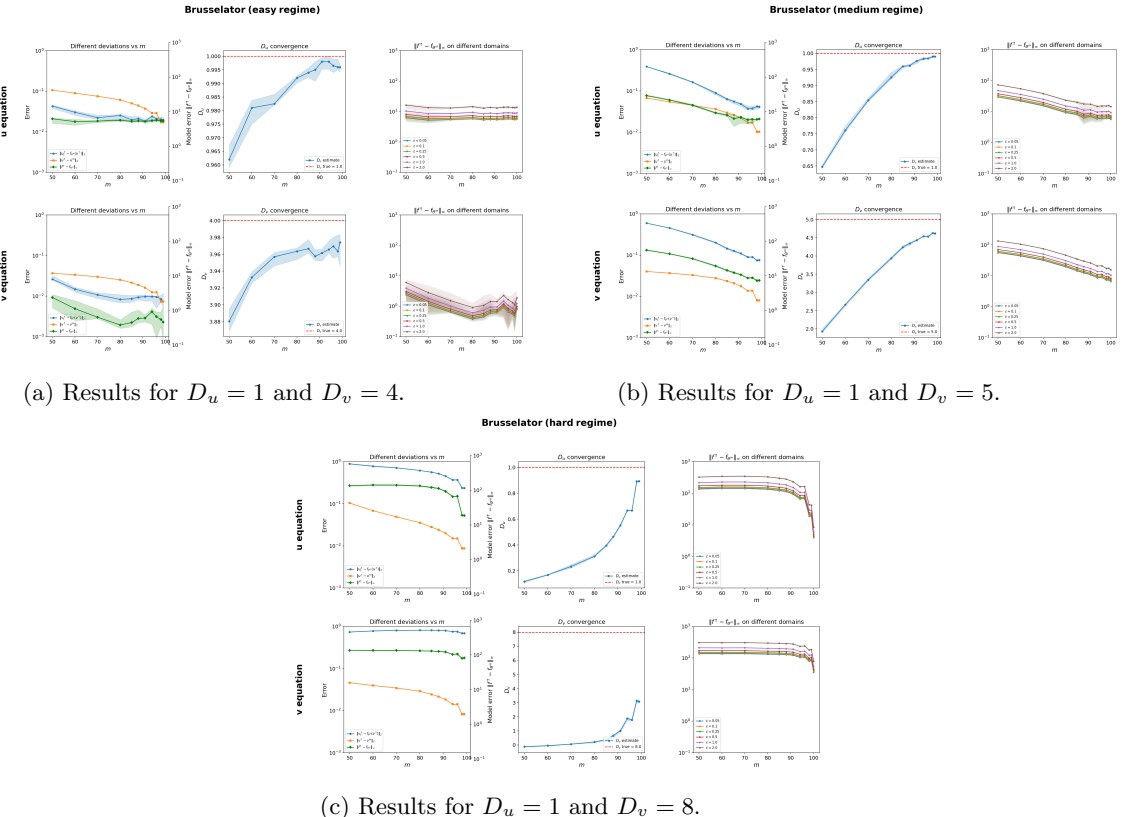

(a) Results for $D_u = 1$ and $D_v = 4$.      (b) Results for $D_u = 1$ and $D_v = 5$.

(c) Results for $D_u = 1$ and $D_v = 8$.

Figure 6: Numerical results for Brusselator with constrained additive structure. Each subplot: $u$-equation (top row), $v$-equation (bottom row). Left: Error metrics. Center: Diffusion coefficient convergence. Right: Model error on $\varepsilon$-neighborhoods.

| Equation | Num. deg. | Den. deg. | Potence | Base fn. | Reps. | Time (s) | Maxiter |
|----------|-----------|-----------|---------|----------|-------|----------|---------|
| $u$ | 2 | 1 | 2 | — | 8 | 45 | 200 |
| $v$ | 2 | 0 | 2 | exp | 8 | 60 | 200 |

Table 6: ParFam architecture for the membrane transport system.

The numerical results are presented in Figure 7. The $v$-equation is consistently recovered with high accuracy, the exponential structure $e^{-u}v$ is identified in virtually every run, with typical coefficients within 2-3% of the true values (e.g., $\hat{f}_v = 0.198\,\Delta v - 0.975\,v\,e^{-0.968\,u} + 0.370$ vs. the true $0.2\,\Delta v - e^{-u}\,v + e^{-1}$). The $u$-equation recovery is more variable, the rational structure $v/(1+u)$ is correctly identified in the best runs (model error $\approx 4$), but some seeds converge to local minima, resulting in larger median errors. This variability is inherent to the non-convex optimization landscape of rational symbolic regression.

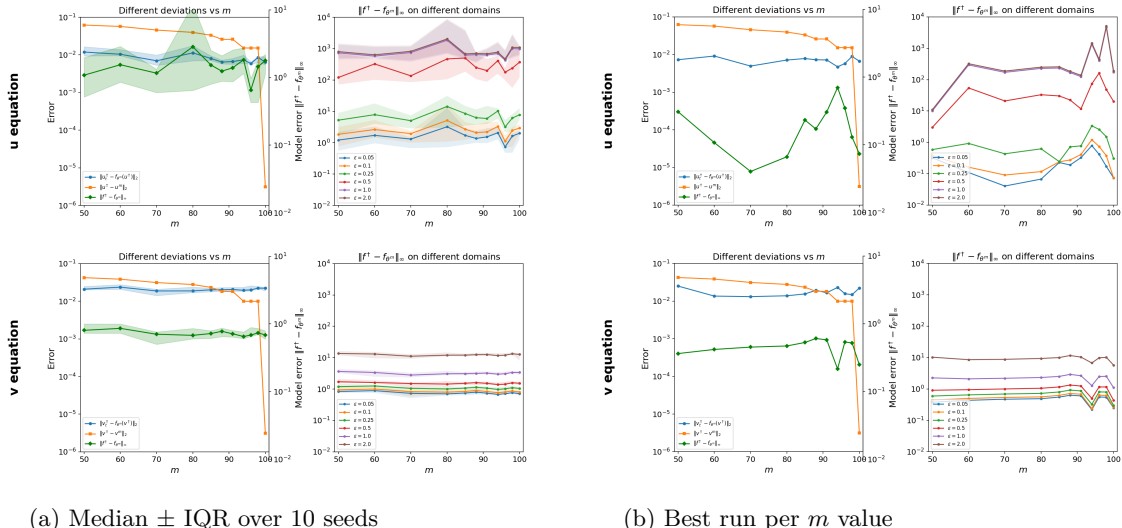

(a) Median $\pm$ IQR over 10 seeds

(b) Best run per $m$ value

Figure 7: Membrane transport results ($D_u = 0.1$, $D_v = 0.2$). (a) Median performance with IQR bands. (b) Best seed per $m$.

## 6  Conclusions

This work focused on investigating the reconstructibility of symbolic (human interpretable) expressions for unknown physical laws corresponding to the right-hand side of a PDE model, as well as reconstructing the unknown state based on noisy and incomplete measurements. We proposed an all-at-once minimization formulation in function space and introduced symbolic networks for approximating the physical law. These networks use rational functions as transformations and base functions as activations. They combine the approximation capabilities of rational functions with the flexibility to represent arithmetic formulas. In addition to reconstructibility results, another key contribution of our work are certain regularity properties of symbolic networks in function space. Our main result addresses the framework where the physical law can be represented by a specific symbolic network architecture. We show in the limit of complete measurements:

1) **State reconstruction:** The state is reconstructed, and the symbolic networks recover an admissible physical law of the PDE model.

2) **Symbolic network representation:** Any limit of such networks remains expressible within the underlying symbolic network architecture, and its parameterization is regularization minimizing. Notably, if the regularization functional is designed to promote sparsity, the proposed approach reconstructs preferentially a simple admissible expression of the physical law.

3) **Unique identifiability:** When the underlying physical law is uniquely identifiable, subject to appropriate conditions, the reconstructed symbolic network in the limit aligns exactly with the unique true physical law.

These theoretical results were supported by first numerical experiments, which provide further confidence in the validity and feasibility of the proposed approach for reconstructing interpretable physical laws.

The theoretical and numerical results of our work provide a foundation for addressing deeper challenges in both symbolic representation and reconstruction of physical laws in PDE models. A promising research direction is to also investigate the case where the physical law cannot be represented within the symbolic network architecture. Even though we briefly outlined an idea for addressing this case, we expect the underlying problem to be significantly more complex. Also interesting are convergence rates and statistical aspects, which are not considered here. Investigating broader designs for symbolic networks could further enrich their applicability. Examples include allowing rational functions with poles or introducing more general base activation functions.

From a practical point of view, it is interesting to extend and deepen the numerical experiments to more general classes of PDEs, which naturally depends on the problem at hand. While the presented experiments demonstrate the feasibility of the approach across ODE and 2D PDE systems with diverse nonlinearities, the current performance remains limited due to the inherent numerical challenges of the problem. We emphasize that the primary contribution is an analysis-guided framework for addressing reconstructibility in symbolic PDE recovery, rather than a fully developed numerical framework. Developing a more comprehensive and robust numerical framework is an important direction for future work.

## Acknowledgement

This research was funded in whole or in part by the Austrian Science Fund (FWF) 10.55776/F100800.

## A   Regularity of symbolic networks

In this section, we provide the proof of the extendability and regularity result stated in Proposition 6 from Subsection 3.4. We first establish the extension property of symbolic networks to function spaces, following (Aarset et al., 2023, Lemma 4).

**Lemma 11.** *Suppose that Assumption 4 holds true. Then the symbolic network $\mathfrak{S}_\sigma^\theta : \mathbb{R}^{n_0^\sigma} \to \mathbb{R}$ induces well-defined Nemytskii operators $\mathfrak{S}_\sigma^\theta : \otimes_{k=0}^\kappa \mathcal{V}_k^\times \to L^q(0,T; L^{\hat{q}}(\Omega))$ and $\mathfrak{S}_\sigma^\theta : \otimes_{k=0}^\kappa \mathcal{V}_k^\times \to \mathcal{W}$, both via $[\mathfrak{S}_\sigma^\theta(u)](t) = \mathfrak{S}_\sigma^\theta(u(t,\cdot))$.*

*Proof.* For fixed $\theta \in \Theta$ we derive continuity of $\mathfrak{S}_\sigma^\theta$ by continuity of the base functions $\sigma_i$ for $1 \leq i \leq L$ (see Definition 1) and continuity of the parameterized rational functions $r_i$ for $1 \leq i \leq L+1$. As a consequence, for $u \in \mathcal{V}$ implying $\mathcal{J}_\kappa u \in L^\infty((0,T) \times \Omega)$ by (3), we obtain that

$$\sup_{0 \leq t \leq T, x \in \Omega} |\mathfrak{S}_\sigma^\theta(t, \mathcal{J}_\kappa u(t,x))| < \infty \tag{10}$$

and hence, the inclusion $\mathfrak{S}_\sigma^\theta(t, \mathcal{J}_\kappa u(t,\cdot)) \in L^{\hat{q}}(\Omega)$ (which is separable) for a.e. $t \in (0,T)$. Now the map $t \mapsto \mathfrak{S}_\sigma^\theta(t, \mathcal{J}_\kappa u(t,\cdot))$ is weakly measurable, which is Lebesgue measurability of $t \mapsto \int_\Omega \mathfrak{S}_\sigma^\theta(t, \mathcal{J}_\kappa u(t,x))w(x)\,\mathrm{d}x$ for all $w \in L^{\hat{q}^*}(\Omega)$. This follows from continuity of $\mathfrak{S}_\sigma^\theta$,

Lebesgue measurability of $w, \mathcal{J}_\kappa u(t, \cdot)$ and due to preservation of measurability under integration. Using Pettis theorem (Roubíček, 2012, Theorem 1.34) we obtain Bochner measurability of $t \mapsto \mathfrak{S}_\sigma^\theta(t, \mathcal{J}_\kappa u(t, \cdot)) \in L^{\hat{q}}(\Omega)$ and with (10) well-definedness of the Nemytskii operator $\mathfrak{S}_\sigma^\theta : \otimes_{k=0}^\kappa \mathcal{V}_k^\times \to L^q(0, T; L^{\hat{q}}(\Omega))$. The remaining assertion follows immediately by $p \geq q$ and $L^{\hat{q}}(\Omega) \hookrightarrow W$. $\qquad \square$

We now turn to the regularity of symbolic networks in function space.

**Rational functions.** For that, we first require the following auxiliary continuity property of rational functions with respect to their parameterization.

**Lemma 12.** *Let $d, n \in \mathbb{N}$ and $U \subset \mathbb{R}^{\binom{n+d}{d}}$ be bounded, then the map*

$$\mathbb{R}^{\binom{n+d}{d}} \times Q(d, n) \ni (a, b) \mapsto \frac{\sum_{0 \leq k_1 + \cdots + k_n \leq d} a_{k_1, \ldots, k_n} \prod_{l=1}^n x_l^{k_l}}{\sum_{0 \leq k_1 + \cdots + k_n \leq d} b_{k_1, \ldots, k_n} \prod_{l=1}^n x_l^{k_l}} \in \mathcal{C}^1(U)$$

*is strong-strong continuous, with the set $Q(d, n)$ given as in Definition 2.*

*Proof.* Let $(a^m, b^m)_{m \in \mathbb{N}} \subset \mathbb{R}^{\binom{n+d}{d}} \times Q(d, n)$ and $(a, b) \in \mathbb{R}^{\binom{n+d}{d}} \times Q(d, n)$ such that $(a^m, b^m) \to (a, b)$ as $m \to \infty$. Denote

$$P^m(x) = \sum_{0 \leq k_1 + \cdots + k_n \leq d} a_{k_1, \ldots, k_n}^m \prod_{l=1}^n x_l^{k_l} \quad \text{and} \quad Q^m(x) = \sum_{0 \leq k_1 + \cdots + k_n \leq d} b_{k_1, \ldots, k_n}^m \prod_{l=1}^n x_l^{k_l}.$$

Similarly, denote by $P, Q$ the polynomials resulting from the coefficients $a$ and $b$, respectively. We aim to show

$$\left\| \frac{P}{Q} - \frac{P^m}{Q^m} \right\|_{\mathcal{C}^1(U)} \to 0 \text{ as } m \to \infty. \tag{11}$$

For that, first note that

$$\left\| \frac{P}{Q} - \frac{P^m}{Q^m} \right\|_{L^\infty(U)} \leq \left\| \frac{1}{Q} \right\|_{L^\infty(U)} \left\| \frac{1}{Q^m} \right\|_{L^\infty(U)} \| PQ^m - P^m Q \|_{L^\infty(U)}. \tag{12}$$

By definition of $Q(d, n)$ (see Definition 2) there exists $\epsilon > 0$ with $Q(x) > \epsilon$ for all $x \in \mathbb{R}^n$. Thus, it holds true that $\|Q^{-1}\|_{L^\infty(U)} \leq 1/\epsilon$. Denote by $M = \max(|x_i| : x \in U, 1 \leq i \leq n) < \infty$. Since $b^m \to b$ as $m \to \infty$, we derive for $x \in U$

$$|Q^m(x) - Q(x)| \leq \sum_{0 \leq k_1 + \cdots + k_n \leq d} |b_{k_1, \ldots, k_n}^m - b_{k_1, \ldots, k_n}| M^{k_1 + \cdots + k_n} \leq C |b^m - b|$$

for some $C > 0$ independent of $m \in \mathbb{N}$. Since this estimation is independent of $x \in U$ we infer that $\|Q^m - Q\|_{L^\infty(U)} \to 0$ as $m \to \infty$. Similarly one can show that $P^m \to P$ in $L^\infty(U)$ as $m \to \infty$. As a consequence, it holds true, using $Q > \epsilon$, for sufficiently large $m \in \mathbb{N}$ with $\|Q^m - Q\|_{L^\infty(U)} \leq \epsilon/2$, that

$$\left\| \frac{1}{Q^m} \right\|_{L^\infty(U)} \leq \left\| \frac{1}{Q - \|Q^m - Q\|_{L^\infty(U)}} \right\|_{L^\infty(U)} < \frac{2}{\epsilon}. \tag{13}$$

Using (13) and that

$$\|PQ^m - P^mQ\|_{L^\infty(U)} \le \|P\|_{L^\infty(U)}\|Q^m - Q\|_{L^\infty(U)} + \|Q\|_{L^\infty(U)}\|P^m - P\|_{L^\infty(U)} \underset{m\to\infty}{\to} 0$$

by the previously discussed convergence of the polynomials $P^m$ to $P$ and $Q^m$ to $Q$, respectively, implies that the the term on the left hand side of (12) converges to 0 as $m \to \infty$. It remains to show that

$$\left\|\nabla\left(\frac{P}{Q} - \frac{P^m}{Q^m}\right)\right\|_{L^\infty(U)} \to 0 \text{ as } m \to \infty.$$

Due to $\nabla(P/Q) = (Q\nabla P - P\nabla Q)/Q^2$ and similarly for $\nabla(P^m/Q^m)$ we obtain

$$\left\|\nabla\left(\frac{P}{Q} - \frac{P^m}{Q^m}\right)\right\|_{L^\infty(U)} \le \left\|\frac{(Q\nabla P - P\nabla Q) - (Q^m\nabla P^m - P^m\nabla Q^m)}{Q^2}\right\|_{L^\infty(U)}$$
$$+ \left\|(Q^m\nabla P^m - P^m\nabla Q^m)\left(\frac{1}{Q^2} - \frac{1}{(Q^m)^2}\right)\right\|_{L^\infty(U)}. \quad (14)$$

We show that both terms converge to zero as $m \to \infty$. As $\|Q^{-2}\|_{L^\infty(U)} \le \epsilon^{-2}$ for the first term, it suffices to show that $Q^m\nabla P^m \to Q\nabla P$ in $L^\infty(U)$ as $m \to \infty$. The convergence $P^m\nabla Q^m \to P\nabla Q$ in $L^\infty(U)$ as $m \to \infty$ follows by a symmetrical argument. By a similar argument as previously, one can show that $\nabla P^m \to \nabla P$ in $L^\infty(U)$ as $m \to \infty$. Together with $Q^m \to Q$ in $L^\infty(U)$ as $m \to \infty$, convergence of the first term on the right hand side of (14) to zero follows. For the second term note that $\|Q^m\nabla P^m - P^m\nabla Q^m\|_{L^\infty(U)} \le \|Q\nabla P - P\nabla Q\|_{L^\infty(U)} + 1$ for sufficiently large $m \in \mathbb{N}$. By (13) for sufficiently large $m$ with $\|Q^m - Q\|_{L^\infty(U)} \le 1$ we have

$$\|Q^{-2} - (Q^m)^{-2}\|_{L^\infty(U)} \le \frac{4}{\epsilon^4}(2\|Q\|_{L^\infty(U)} + 1)\|Q^m - Q\|_{L^\infty(U)},$$

which converges to zero as $m \to \infty$, proving convergence of (14) to zero and finally, the claim in (11). This concludes the assertions of the lemma. $\qquad\square$

**Symbolic Networks.** We address the regularity statements in Proposition 6 following a similar strategy as in the proof of (Holler & Morina, 2024a, Lemma 17) based on Lemma 12.

**Lemma 13** (Weak-strong continuity of $\mathfrak{S}$). *Suppose that Assumption 4 holds true and that the base functions $(\sigma_i)_{1\le i \le L}$ are locally Lipschitz continuous, then*

$$\mathfrak{S} : \Theta \times \mathcal{V} \to \mathcal{W}$$
$$(\theta, v) \mapsto \mathfrak{S}(\theta, v) =: \mathfrak{S}_\sigma^\theta(\mathcal{J}_\kappa v)$$

*is weak-strong continuous.*

*Proof.* We follow a similar strategy as in the proof of (Holler & Morina, 2024a, Lemma 17). Let $(\theta^m, u^m) \rightharpoonup (\theta, u) \in \Theta \times \mathcal{V}$ weakly as $m \to \infty$. We prove that $\mathfrak{S}(\theta^m, u^m) \to \mathfrak{S}(\theta, u)$ in $\mathcal{W}$ as $m \to \infty$. Since $\Theta$ is finite dimensional, the convergence of the parameters $(\theta^m)_m$ holds in the strong sense. The weak convergence of $(u^m)_m$ in $\mathcal{V}$ implies strong convergence in $L^p(0, T; W^{\kappa, \hat{p}}(\Omega))$, which can be seen as follows. In case $W^{\kappa, \hat{p}}(\Omega) \hookrightarrow V$ the Aubin-Lions Lemma (Roubíček, 2012, Lemma 7.7) implies

$$\mathcal{V} = L^p(0, T; V) \cap W^{1,p,p}(0, T; \tilde{V}) \hookrightarrow\hspace{-0.5em}\rightarrow L^p(0, T; W^{\kappa, \hat{p}}(\Omega)).$$

Otherwise for $\tilde{V} \hookrightarrow W^{\kappa,\hat{p}}(\Omega)$ above embedding follows again by the Aubin-Lions lemma, since in this case $\mathcal{V} \subseteq L^p(0,T;V) \cap W^{1,p,p}(0,T;W^{\kappa,\hat{p}}(\Omega))$. With this, applying (Holler & Morina, 2024a, Lemma 31) we derive that $\mathcal{J}_\kappa u^m \to \mathcal{J}_\kappa u$ strongly in $\otimes_{k=0}^\kappa L^p(0,T;L^{\hat{p}}(\Omega)^{p_k})$ as $m \to \infty$. Another important conclusion that can be drawn from the weak convergence of $(u^m)_m$ in $\mathcal{V}$ is boundedness of $(\mathcal{J}_\kappa u^m)_m$ due to (3). Thus, there exists an origin-centered ball $\mathcal{B}_{M_0}(0) \subseteq \mathbb{R}^{1+\sum_{k=0}^\kappa p_k}$ of radius $M_0 > 0$ such that $(t, \mathcal{J}_\kappa u^m(t,x)) \in U$ for a.e. $(t,x) \in (0,T) \times \Omega$ for all $m \in \mathbb{N}$. In the following we denote by $z = \mathcal{J}_\kappa u, z^m = \mathcal{J}_\kappa u^m \in \otimes_{k=0}^\kappa L^p(0,T;L^{\hat{p}}(\Omega)^{p_k})$ for $m \in \mathbb{N}$ and show first

$$\mathfrak{S}_\sigma^{\theta^m}(z^m) \to \mathfrak{S}_\sigma^\theta(z) \quad \text{in } L^q(0,T;L^{\hat{q}}(\Omega)) \quad \text{as } m \to \infty. \tag{15}$$

Omitting the notational dependence of $z, (z^m)_m$ on time and space we estimate for a.e. $(t,x) \in (0,T) \times \Omega$ pointwise

$$|\mathfrak{S}_\sigma^{\theta^m}(t,z^m) - \mathfrak{S}_\sigma^\theta(t,z)| \le \underbrace{|\mathfrak{S}_\sigma^{\theta^m}(t,z^m) - \mathfrak{S}_\sigma^{\theta^m}(t,z)|}_{\text{(I)}} + \underbrace{|\mathfrak{S}_\sigma^{\theta^m}(t,z) - \mathfrak{S}_\sigma^\theta(t,z)|}_{\text{(II)}}. \tag{16}$$

**Estimation of (I).** We recall that for $\theta^m$ parameterizing the rational functions $r_i^m$ (and $\theta$ parameterizing $r_i$) for $1 \le i \le L+1$ we have for $(t,y) \in \mathbb{R}^{1+\sum_{k=0}^\kappa p_k} = \mathbb{R}^{n_0^\sigma}$

$$\mathfrak{S}_\sigma^{\theta^m} : \mathbb{R}^{n_0^\sigma} \to \mathbb{R}$$
$$(t,y) \mapsto r_{L+1}^m((\sigma_L \circ r_L^m \circ \cdots \circ \sigma_1 \circ r_1^m)(t,y), (t,y)).$$

For $1 \le i \le L$ we define iteratively origin-centered balls $\mathcal{B}_{M_i}(0) \subseteq \mathbb{R}^{n_i^\sigma}$ as follows. We fix $\epsilon > 0$. As the rational function $r_i$ is continuous, the image $r_i(\mathcal{B}_{M_{i-1}}(0))$ is bounded and included in $\mathcal{B}_{\tilde{M}_i}(0) \subseteq \mathbb{R}^{n_i^r}$ for some $\tilde{M}_i > 0$. By Lemma 12, for sufficiently large $m \in \mathbb{N}$, it holds true that $r_i^m(\mathcal{B}_{M_{i-1}}(0)) \subseteq \mathcal{B}_{(1+\epsilon)\tilde{M}_i}(0)$. Similarly, by continuity of $\sigma_i$ there exists some $M_i > 0$ with $\sigma_i(\mathcal{B}_{(1+\epsilon)\tilde{M}_i}(0)) \subseteq \mathcal{B}_{M_i}(0) \subseteq \mathbb{R}^{n_i^\sigma}$.

Thus, with $U_i := \mathcal{B}_{M_i}(0) \subseteq \mathbb{R}^{n_i^\sigma}$ and $V_i := \mathcal{B}_{(1+\epsilon)\tilde{M}_i}(0) \subseteq \mathbb{R}^{n_i^r}$ for $0 \le i \le L$, we derive that $r_i^m(U_{i-1}), r_i(U_{i-1}) \subseteq V_i$ and $\sigma_i(V_i) \subseteq U_i$ for $1 \le i \le L$. Furthermore, it holds true that $z^m(t,x), z(t,x) \in U_0$ for a.e. $(t,x) \in (0,T) \times \Omega$.

Now as $r_i \in \mathcal{C}^1(U_{i-1})$ it is Lipschitz continuous on $U_{i-1}$ with some constant $\mathcal{L}_i^r > 0$ (indeed one can choose $\mathcal{L}_i^r = |r_i|_{\mathcal{C}^1(U_{i-1})}$) for $1 \le i \le L+1$. Employing Lemma 12, we infer that for sufficiently large $m \in \mathbb{N}$ also the $r_i^m$ are Lipschitz continuous on $U_{i-1}$ with Lipschitz constant $\mathcal{L}_i^r + \epsilon$ for $1 \le i \le L+1$. Using local Lipschitz continuity of the $\sigma_i$, we obtain Lipschitz continuity of the $\sigma_i$ on $V_i$ with constant $\mathcal{L}_i^\sigma > 0$ for $1 \le i \le L$. As a consequence, we can estimate the term in (I) by

$$|\mathfrak{S}_\sigma^{\theta^m}(t,z^m) - \mathfrak{S}_\sigma^{\theta^m}(t,z)| \le \underbrace{(\mathcal{L}_{L+1}^r + \epsilon)(1 + \prod_{i=1}^L (\mathcal{L}_i^r + \epsilon)\mathcal{L}_i^\sigma)}_{=:\mathcal{L}} |z^m - z| \tag{17}$$

**Estimation of (II).** Assume w.l.o.g. that the Lipschitz constants fulfill $\mathcal{L}_i^r, \mathcal{L}_i^\sigma \ge 1$. Again employing Lemma 12 we can choose $m \in \mathbb{N}$ sufficiently large such that

$$\|r_i - r_i^m\|_{L^\infty(U_{i-1})} < \epsilon \tag{18}$$

for $1 \le i \le L+1$. We define the auxiliary networks $(\mathcal{S}_i)_{i=0}^{L+1}$ as follows

$$\mathcal{S}_0(\theta^m, \theta, \cdot) = \mathfrak{S}(\theta^m, \cdot), \quad \mathcal{S}_{L+1}(\theta^m, \theta, \cdot) = \mathfrak{S}(\theta, \cdot) \quad \text{and for } 1 \le s \le L$$

$$\mathcal{S}_s(\theta^m, \theta, \cdot) = r_{L+1}((\sigma_L \circ r_L \circ \cdots \circ \sigma_{L-s+2} \circ r_{L-s+2} \circ \sigma_{L-s+1} \circ r_{L-s+1}^m \circ \cdots \circ \sigma_1 \circ r_1^m)(\cdot), \cdot).$$

Note first that due to the considerations on the estimation of (I) (with $0 \in U_0$) it follows that there exists some $C > 0$ such that for sufficiently large $m \in \mathbb{N}$

$$(\sigma_s \circ r_s^m \circ \cdots \circ \sigma_1 \circ r_1^m)(0) < C$$

for $1 \le s \le L$. We estimate the term in (II) by the telescope sum

$$|\mathfrak{S}_\sigma^{\theta^m}(t,z) - \mathfrak{S}_\sigma^{\theta}(t,z)| \le \sum_{s=0}^{L} |\mathcal{S}_{s+1}(\theta^m, \theta, t, z) - \mathcal{S}_s(\theta^m, \theta, t, z)|. \tag{19}$$

Using the Lipschitz constants derived in view of the estimation of (I) we obtain for sufficiently large $m \in \mathbb{N}$, defining $\mathcal{T}_s^m = \sigma_s \circ r_s^m \circ \cdots \circ \sigma_1 \circ r_1^m$, that

$$|\mathcal{S}_{s+1}(\theta^m, \theta, t, z) - \mathcal{S}_s(\theta^m, \theta, t, z)| \tag{20}$$

$$\le (\mathcal{L}_{L+1}^r + \epsilon) \prod_{i=L-s+2}^{L} (\mathcal{L}_i^r + \epsilon) \mathcal{L}_i^\sigma |(\sigma_{L-s+1} \circ r_{L-s+1}) - (\sigma_{L-s+1} \circ r_{L-s+1}^m)|(\mathcal{T}_{L-s}^m(t,z)).$$

Since both $r_{L-s+1}(\mathcal{T}_{L-s}^m(t,z)), r_{L-s+1}^m(\mathcal{T}_{L-s}^m(t,z)) \in V_{L-s+1}$ and $\mathcal{T}_{L-s}^m(t,z) \in U_{L-s}$ we can estimate using (18)

$$|(\sigma_{L-s+1} \circ r_{L-s+1}) - (\sigma_{L-s+1} \circ r_{L-s+1}^m)|(\mathcal{T}_{L-s}^m(t,z))$$
$$\le \mathcal{L}_{L-s+1}^\sigma |r_{L-s+1}(\mathcal{T}_{L-s}^m(t,z)) - r_{L-s+1}^m(\mathcal{T}_{L-s}^m(t,z))| \le \epsilon \mathcal{L}_{L-s+1}^\sigma. \tag{21}$$

Combining (19),(20),(21) together with the Lipschitz constants assumed to be larger than one, we conclude that

$$|\mathfrak{S}_\sigma^{\theta^m}(t,z) - \mathfrak{S}_\sigma^\theta(t,z)| \le \epsilon L \mathcal{L}. \tag{22}$$

Finally, by (16), (17) and (22) we derive the pointwise estimate

$$|\mathfrak{S}_\sigma^{\theta^m}(t, z^m) - \mathfrak{S}_\sigma^\theta(t, z)| \le \mathcal{L}|z^m - z| + \epsilon L \mathcal{L}. \tag{23}$$

**Convergence in function space.** In view of (15) we derive using (23) and the triangle inequality that

$$\|\mathfrak{S}_\sigma^{\theta^m}(z^m) - \mathfrak{S}_\sigma^\theta(z)\|_{L^q(0,T;L^{\hat{q}}(\Omega))} \le \mathcal{L}\|z^m - z\|_{L^q(0,T;L^{\hat{q}}(\Omega))} + \epsilon L \mathcal{L} T^{1/q} |\Omega|^{1/\hat{q}}. \tag{24}$$

Employing Hölder's inequality, we can estimate for some generic constant $C > 0$

$$\|z^m - z\|_{L^q(0,T;L^{\hat{q}}(\Omega))} \le C\|\mathcal{J}_\kappa u^m - \mathcal{J}_\kappa u\|_{\otimes L^p(0,T;L^{\hat{p}}(\Omega)^{p_k})} \le C\|u^m - u\|_{L^p(0,T;W^{\kappa,\hat{p}}(\Omega))} \tag{25}$$

where the last inequality follows by definition of the differential operator $\mathcal{J}_\kappa$. Since $u^m \to u$ in $L^p(0,T;W^{\kappa,\hat{p}}(\Omega))$ as $m \to \infty$ and $\epsilon$ can be chosen arbitrarily small (with resulting larger $m$ to fulfill underlying inequalities), we conclude by (24) and (25) that (15) holds true. With the embedding $L^q(0,T;L^{\hat{q}}(\Omega)) \hookrightarrow \mathcal{W}$ this implies that

$$\mathfrak{S}(\theta^m, u^m) \to \mathfrak{S}(\theta, u) \text{ as } m \to \infty \text{ in } \mathcal{W},$$

proving the claimed weak-strong continuity of the joint operator $\mathfrak{S}$. $\qquad \square$

Following the proof of Lemma 13 we can extract the following regularity property.

**Corollary 14.** *Suppose that Assumption 4 holds true and that the base functions $(\sigma_i)_{1 \leq i \leq L}$ are locally Lipschitz continuous. Then for bounded $U \subset \mathbb{R}^{n_0^\sigma}$ the map*

$$\Theta \ni \theta \to \mathfrak{S}_\sigma^\theta \in L^\infty(U)$$

*is continuous.*

*Proof.* Let $(\theta^m)_m \subset \Theta$ with $\theta^m \to \theta \in \Theta$ as $m \to \infty$. Following the proof of Lemma 13 we derive by (23) that for every $\epsilon > 0$ the estimation

$$|\mathfrak{S}_\sigma^{\theta^m}(w) - \mathfrak{S}_\sigma^\theta(w)| \leq \epsilon L \mathcal{L}$$

holds true for sufficiently large $m \in \mathbb{N}$ for all $w \in U$, proving the assertion. $\qquad\square$

Combining Corollary 14 with Lemmata 8 and 13 completes the proof of the statement in Proposition 6. Note that a crucial key property to derive Lemma 13 and, consequently, Proposition 6, apart from Lemma 8, is the uniform state space regularity assumption in (3). This raises the question whether one can avoid the underlying embedding under stronger regularity assumptions on the activation base functions $(\sigma_i)_{1 \leq i \leq L}$. An alternative approach is presented e.g., in (Holler & Morina, 2024a, Lemma 17), where global Lipschitz continuity of the activation functions is sufficient, rather than local Lipschitz continuity, in the case of affine linear transformations. However, this result does not cover rational transformations, which are the focus here. Some initial considerations addressing this question are outlined below.

**Remark 15.** *We conjecture that the regularity assumption (3) can be avoided. For the special case of rational transformations $r = p/q$ with $\deg(p) \geq \deg(q) \geq \deg(p) - 1$ and $q > 0$, already covering a large class of rational functions with rather strong approximation properties (see Boullé et al. (2020) and Holler & Morina (2025a)), one can argue w.l.o.g. in one dimension as follows. Rational functions of the type above behave like affine linear functions towards $\pm\infty$ and their seminorm fulfills $|r|_{\mathcal{C}^1(\mathbb{R})} < \infty$. Furthermore, for $r$ parameterized by coefficients $\theta \in \Theta$ and coefficients $\theta^m$ parameterizing rationals $r^m = p^m/q^m$ (fulfilling $\deg(p^m) \geq \deg(q^m) \geq \deg(p^m) - 1$ and $q^m > 0$) with $\theta^m \to \theta$ as $m \to \infty$ it follows that $|r^m|_{\mathcal{C}^1(\mathbb{R})} \to |r|_{\mathcal{C}^1(\mathbb{R})}$ as $m \to \infty$. In other words $r$ and $(r^m)_m$ are jointly globally Lipschitz continuous with the same constant. With this, an analogous result as in (Holler & Morina, 2024a, Lemma 17) can be obtained following its proof in combination with the one of Lemma 13. If the rational transformations are general polynomials of degree $d$, we conjecture that (3) can be weakened to the regularity assumption that the state space is of the form that $(\mathcal{J}_\kappa u^m)_m$ converges in space in $L^{\hat{p}\rho_0}(\Omega)$ and is bounded in $L^{(d-1)^i\hat{p}\rho_i}(\Omega)$ for the layer index $1 \leq i \leq L$ where $\sum_{i=0}^L \rho_i^{-1} = 1$. We expect a similar result for general rational transformations $r = p/q$ for polynomials $p, q$ with $\deg(p) - \deg(q)$ instead of $d$. These regularity assumptions are obviously more difficult to fulfill the deeper the network is and the more complex the rational transformations are.*

## B    Proofs of identification results

In the following we provide the detailed arguments of the assertions in Section 4, starting with the proof of Lemma 8.

*Proof of Lemma 8.* The assertion essentially follows from the *direct method*. Since the regularization $\mathcal{R}$ is proper, there exists an infimizing sequence $(u_k, \theta_k)_k \subset \mathcal{V} \times \Theta$ of problem $(\mathcal{P}^m)$. Due to coercivity of $\mathcal{R}$, $(u_k)_k$ is bounded in $\mathcal{V}$ and $(\theta_k)_k$ in $\Theta$. As a consequence, due to reflexivity of $\mathcal{V}$, the sequence $(u_k)_k$ admits a weakly convergent subsequence in $\mathcal{V}$ with limit $\hat{u} \in \mathcal{V}$, and $(\theta_k)_k$ a strongly convergent subsequence in $\Theta$ with limit $\hat{\theta} \in \Theta$, since $\Theta$ is finite dimensional (w.l.o.g. for the entire sequences) and closed by design (see Subsection 3.2). We derive

$$\mathfrak{S}_\sigma^{\theta_k}(\mathcal{J}_\kappa u_k) \to \mathfrak{S}_\sigma^{\hat{\theta}}(\mathcal{J}_\kappa \hat{u}) \quad \text{in } \mathcal{W} \quad \text{as } k \to \infty$$

using Proposition 6. Now since $\partial_t u_k \rightharpoonup \partial_t \hat{u}$ in $L^p(0, T; \tilde{V})$ as $k \to \infty$ the weak convergence also holds in $\mathcal{W}$ as $L^p(0, T; \tilde{V}) \hookrightarrow \mathcal{W}$ by $\tilde{V} \hookrightarrow W$. Furthermore, weak-weak continuity of $K^m$ implies $K^m u_k \rightharpoonup K^m \hat{u}$ in $\mathcal{Y}$ as $k \to \infty$. Combining these convergences with weak lower semicontinuity of $\lambda^m \|\cdot\|_{\mathcal{W}}^q$, $\mu^m \|\cdot\|_{\mathcal{Y}}^r$ and the regularization $\mathcal{R}$ in the respective spaces, it follows that $(\hat{u}, \hat{\theta})$ solves $(\mathcal{P}^m)$. $\qquad \square$

We conclude this section by providing the proof of our main result, Theorem 9.

*Proof of Theorem 9.* Since $f^\dagger$ is representable, there exists $\theta^\dagger \in \Theta$ such that $f^\dagger = \mathfrak{S}_\sigma^{\theta^\dagger}$. We estimate the objective functional of $(\mathcal{P}^m)$ by

$$\lambda^m \|\partial_t u^m - \mathfrak{S}_\sigma^{\theta^m}(\mathcal{J}_\kappa u^m)\|_{\mathcal{W}}^q + \mu^m \|K^m u^m - y^m\|_{\mathcal{Y}}^r + \mathcal{R}(u^m, \theta^m)$$
$$\leq \lambda^m \|\partial_t u^\dagger - \mathfrak{S}_\sigma^{\theta^\dagger}(\mathcal{J}_\kappa u^\dagger)\|_{\mathcal{W}}^q + \mu^m \|K^m u^\dagger - y^m\|_{\mathcal{Y}}^r + \mathcal{R}(u^\dagger, \theta^\dagger). \quad (26)$$

As $\partial_t u^\dagger = \mathfrak{S}_\sigma^{\theta^\dagger}(\mathcal{J}_\kappa u^\dagger)$ and $\|K^m u^\dagger - y^m\|_{\mathcal{Y}}^r = \delta(m)$, the right hand side of (26) converges to $\mathcal{R}(u^\dagger, \theta^\dagger)$ as $m \to \infty$. This implies boundedness of $(\|u^m\|_{\mathcal{V}})_m$ and $(\|\theta^m\|)_m$ by coercivity of $\mathcal{R}$. As a consequence, there exists a weakly convergent subsequence of the $(u^m)_m$ with limit $\tilde{u}$ in $\mathcal{V}$. We denote it w.l.o.g. by the original indices as we will show $u^m \rightharpoonup u^\dagger$ in $\mathcal{V}$ as $m \to \infty$. Since $\lim_{m \to \infty} \mu_m = \infty$ the convergence $\lim_{m \to \infty} \|K^m u^m - y^m\|_{\mathcal{Y}} = 0$ follows. We estimate

$$\|K^\dagger \tilde{u} - K^\dagger u^\dagger\|_{\mathcal{Y}} \leq \|K^\dagger \tilde{u} - K^\dagger u^m\|_{\mathcal{Y}} + \|K^\dagger u^m - K^m u^m\|_{\mathcal{Y}}$$
$$+ \|K^m u^m - K^m u^\dagger\|_{\mathcal{Y}} + \|K^m u^\dagger - K^\dagger u^\dagger\|_{\mathcal{Y}}.$$

The first term converges to zero by weak-strong continuity of $K^\dagger$. The second and fourth term converge to zero by (5), since $(u^m)_m$ and the constant sequence $(u^\dagger)_m$ are weakly convergent. The third term converges to zero by (4), since $K^m u^\dagger = y^m$ and $\lim_{m \to \infty} \delta(m) = 0$. Thus, we conclude that $K^\dagger \tilde{u} = K^\dagger u^\dagger$ and finally, $\tilde{u} = u^\dagger$ by injectivity of $K^\dagger$. Since $(\|\theta^m\|)_m$ is bounded, there exists a convergent subsequence $(\theta^{m_l})_l$ with limit $\tilde{\theta} \in \Theta$ by closedness of $\Theta$. Using $\lambda^m \to \infty$ as $m \to \infty$ we derive

$$\lim_{l \to \infty} \|\partial_t u^{m_l} - \mathfrak{S}_\sigma^{\theta^{m_l}}(\mathcal{J}_\kappa u^{m_l})\|_{\mathcal{W}} = 0 \quad (27)$$

and with $\partial_t u^{m_l} \rightharpoonup \partial_t u^\dagger$, as in the proof of Lemma 8, boundedness of $(\partial_t u^{m_l})_{m_l}$. As a consequence, also $(\mathfrak{S}_\sigma^{\theta^{m_l}}(\mathcal{J}_\kappa u^{m_l}))_{m_l}$ is bounded and there exists $g \in \mathcal{W}$ such that $\mathfrak{S}_\sigma^{\theta^{m_l}}(\mathcal{J}_\kappa u^{m_l}) \rightharpoonup g$ in $\mathcal{W}$ as $l \to \infty$ (w.l.o.g. for the entire sequence). Employing weak lower semicontinuity of the $\|\cdot\|_{\mathcal{W}}$-norm

in (27) yields $g = \partial_t u^\dagger$. Using Proposition 6 we obtain that $u^\dagger$ and the reconstructed physical law $\mathfrak{S}_\sigma^{\tilde{\theta}}$ fulfill

$$\partial_t u^\dagger = \mathfrak{S}_\sigma^{\tilde{\theta}}(\mathcal{J}_\kappa u^\dagger). \tag{28}$$

The assertion on $L^\infty_{\text{loc}}(\mathbb{R}^{n_0^\sigma})$-convergence follows directly by Proposition 6. Finally, for any solution $(u^\dagger, \theta^\dagger)$ of $\partial_t u^\dagger = \mathfrak{S}_\sigma^{\theta^\dagger}$ it follows by (26) that

$$\liminf_{m \to \infty} \mathcal{R}(u^m, \theta^m) \leq \mathcal{R}(u^\dagger, \theta^\dagger)$$

which by weak lower semicontinuity of $\mathcal{R}$ implies $\mathcal{R}(u^\dagger, \tilde{\theta}) \leq \mathcal{R}(u^\dagger, \theta^\dagger)$. $\qquad\square$

As a consequence of Theorem 9 the result in Corollary 10 holds.

*Proof of Corollary 10.* Following Theorem 9, we conclude from (28) that

$$\mathfrak{S}_\sigma^{\tilde{\theta}}(\mathcal{J}_\kappa u^\dagger) = \partial_t u^\dagger = \mathfrak{S}_\sigma^{\theta^\dagger}(\mathcal{J}_\kappa u^\dagger).$$

Using the identifiability condition $(\mathcal{I})$, we deduce that $\mathfrak{S}_\sigma^{\tilde{\theta}} = \mathfrak{S}_\sigma^{\theta^\dagger}$. Since this equality is independent of the convergent subsequence of $(\theta^m)_m$, together with the result of Theorem 9, we conclude, as claimed that

$$\mathfrak{S}_\sigma^{\theta^m} \to \mathfrak{S}_\sigma^{\theta^\dagger} = f^\dagger \quad \text{in } L^\infty_{\text{loc}}(\mathbb{R}^{n_0^\sigma}) \text{ as } m \to \infty. \qquad\square$$

## C  Convergence condition for sampling operators

In this section we provide a proof of the convergence condition (5) for the measurement operators defined in Subsection 5.1. Recall the full measurement operator

$$K^\dagger : \mathcal{V} \to \mathcal{Y}, \quad [K^\dagger u](t) = \sum_{i \in \mathbb{N}} \langle e_i, u(t) \rangle e_i \tag{29}$$

for $u \in \mathcal{V}$ and $t \in [0, T]$ with $(e_i)_{i \in \mathbb{N}}$ a fixed orthonormal system of $L^2(\Omega)$. Note that $K^\dagger u$ is Bochner measurable for $u \in \mathcal{V}$ by Pettis theorem since $L^2(\Omega)$ is separable and for $w \in L^2(\Omega)$ the map $[0, T] \ni t \mapsto \langle [K^\dagger u](t), w \rangle = \langle w, u(t) \rangle$ is Lebesgue measurable (by Bochner measurability of $u$). Well-definedness follows from Bessel's inequality. The reduced measurement operators are given for $1 \leq j \leq m-1$ and $t \in [t_j, t_{j+1})$ or $j = m$ and $t \in [t_m, t_{m+1}]$ by

$$K^m : \mathcal{V} \to \mathcal{Y}, \quad [K^m u](t) = \sum_{i=1}^m \left( \Delta_m^{-1} \int_{t_j}^{t_{j+1}} \langle e_i, u(s) \rangle \, ds \right) e_i \tag{30}$$

for $u \in \mathcal{V}$. Here $0 = t_1 < t_2 < \cdots < t_m < t_{m+1} = T$ is the $m$-equidistant grid on $[0, T]$ for $m \in [0, T]$ and $\Delta_m := T/m$. Bochner measurability follows again by Pettis theorem since $[0, T] \ni t \mapsto \langle [K^m u](t), w \rangle$ is a step function for $w \in L^2(\Omega)$. We will verify shortly that the operator in (30) is in fact well defined. Now since $K^m$ is linear, weak-weak continuity is equivalent to continuity. Since $(e_i)_i$ is an orthonormal system we derive for $t \in [t_j, t_{j+1}]$ that

$$\|[K^m u](t)\|_{L^2(\Omega)}^2 = \Delta_m^{-2} \sum_{i=1}^m \left( \int_{t_j}^{t_{j+1}} \langle e_i, u(s) \rangle \, ds \right)^2.$$

As a consequence, for $u \in \mathcal{V}$ it holds

$$\|K^m u\|_{\mathcal{Y}}^2 = \int_0^T \|[K^m u](t)\|_{L^2(\Omega)}^2 \, \mathrm{d}t = \sum_{j=1}^m \Delta_m \Delta_m^{-2} \sum_{i=1}^m \left( \int_{t_j}^{t_{j+1}} \langle e_i, u(s) \rangle \, \mathrm{d}s \right)^2$$

which due to Hölder's inequality using $\Delta_m = t_{j+1} - t_j$ implies that

$$\|K^m u\|_{\mathcal{Y}}^2 \leq \sum_{j=1}^m \Delta_m^{-1} \sum_{i=1}^m \Delta_m \int_{t_j}^{t_{j+1}} |\langle e_i, u(s) \rangle|^2 \, \mathrm{d}s = \sum_{j=1}^m \int_{t_j}^{t_{j+1}} \left( \sum_{i=1}^m |\langle e_i, u(s) \rangle|^2 \right) \mathrm{d}s.$$

By employing Bessel's inequality and the embedding $\mathcal{V} \hookrightarrow \mathcal{Y}$ we derive

$$\|K^m u\|_{\mathcal{Y}}^2 \leq \sum_{j=1}^m \int_{t_j}^{t_{j+1}} \|u(s)\|_{L^2(\Omega)}^2 \, \mathrm{d}s = \int_0^T \|u(s)\|_{L^2(\Omega)}^2 \, \mathrm{d}s = \|u\|_{\mathcal{Y}}^2 \leq c \|u\|_{\mathcal{V}}^2$$

for some suitable $c > 0$ proving continuity of $K^m$ and more importantly well definedness of the operator $K^m$. It remains to show (5) that $K^m u^m - K^\dagger u^m \to 0$ in $\mathcal{Y}$ as $m \to \infty$ for any weakly convergent sequence $(u^m)_m \subset \mathcal{V}$. For that, let $u^m \rightharpoonup u$ in $\mathcal{V}$ as $m \to \infty$. Then, similar transformations as above yield

$$\|K^m u^m - K^\dagger u^m\|_{\mathcal{Y}}^2 = \sum_{j=1}^m \int_{t_j}^{t_{j+1}} \left\| \sum_{i=1}^m \left( \Delta_m^{-1} \int_{t_j}^{t_{j+1}} \langle e_i, u^m(s) \rangle \, \mathrm{d}s \right) e_i - \sum_{i \in \mathbb{N}} \langle e_i, u^m(t) \rangle e_i \right\|_{L^2(\Omega)}^2 \mathrm{d}t.$$

Again using that $(e_i)_i$ is an orthonormal system gives

$$\sum_{j=1}^m \int_{t_j}^{t_{j+1}} \left\| \sum_{i=1}^m \left( \Delta_m^{-1} \int_{t_j}^{t_{j+1}} \langle e_i, u^m(s) \rangle \, \mathrm{d}s \right) e_i - \sum_{i \in \mathbb{N}} \langle e_i, u^m(t) \rangle e_i \right\|_{L^2(\Omega)}^2 \mathrm{d}t$$

$$= \sum_{j=1}^m \int_{t_j}^{t_{j+1}} \left\| \sum_{i=1}^m \left( \langle e_i, u^m(t) \rangle - \Delta_m^{-1} \int_{t_j}^{t_{j+1}} \langle e_i, u^m(s) \rangle \, \mathrm{d}s \right) e_i \right\|_{L^2(\Omega)}^2 \mathrm{d}t$$

$$+ \sum_{j=1}^m \int_{t_j}^{t_{j+1}} \left\| \sum_{i \geq m+1} \langle e_i, u^m(t) \rangle e_i \right\|_{L^2(\Omega)}^2 \mathrm{d}t. \quad (31)$$

We argue that the right-hand side of (31) converges to zero as $m \to \infty$. For the second term on the right-hand side of (31) we derive by previous arguments

$$\sum_{j=1}^m \int_{t_j}^{t_{j+1}} \left\| \sum_{i \geq m+1} \langle e_i, u^m(t) \rangle e_i \right\|_{L^2(\Omega)}^2 \mathrm{d}t = \int_0^T \left\| \sum_{i \geq m+1} \langle e_i, u^m(t) \rangle e_i \right\|_{L^2(\Omega)}^2 \mathrm{d}t$$

$$\leq 2 \int_0^T \left\| \sum_{i \geq m+1} \langle e_i, u^m(t) - u(t) \rangle e_i \right\|_{L^2(\Omega)}^2 \mathrm{d}t + 2 \int_0^T \left\| \sum_{i \geq m+1} \langle e_i, u(t) \rangle e_i \right\|_{L^2(\Omega)}^2 \mathrm{d}t$$

$$= \underbrace{2 \int_0^T \sum_{i \geq m+1} |\langle e_i, u^m(t) - u(t) \rangle|^2 \, \mathrm{d}t}_{=:I} + \underbrace{2 \int_0^T \sum_{i \geq m+1} |\langle e_i, u(t) \rangle|^2 \, \mathrm{d}t}_{=:II}.$$

The term I can be estimated using Bessel's inequality by

$$\mathrm{I} \leq \int_0^T \|u^m(t) - u(t)\|^2_{L^2(\Omega)} \, \mathrm{d}t = \|u^m - u\|^2_{\mathcal{Y}}$$

which converges to zero as $m \to \infty$ due to the compact embedding $\mathcal{V} \hookrightarrow\hookrightarrow \mathcal{Y}$. For term II note that due to Bessel's inequality, $t \mapsto \sum_{i \geq m+1} |\langle e_i, u(t)\rangle|^2$ is majorized by $t \mapsto \|u(t)\|^2_{L^2(\Omega)}$ which is integrable on $[0, T]$ by $\mathcal{V} \hookrightarrow \mathcal{Y}$. As a consequence, since $\sum_{i \in \mathbb{N}} |\langle e_i, u(t)\rangle|^2$ is convergent by Parseval's identity, its tails converge to zero, such that with Lebesgue's dominated convergence we recover convergence of the term II to zero as $m \to \infty$. It remains to verify that the first term on the right-hand side of (31) converges to zero as $m \to \infty$. It can be rewritten by

$$\sum_{j=1}^m \int_{t_j}^{t_{j+1}} \sum_{i=1}^m \left( \langle e_i, u^m(t)\rangle - \Delta_m^{-1} \int_{t_j}^{t_{j+1}} \langle e_i, u^m(s)\rangle \, \mathrm{d}s \right)^2 \mathrm{d}t$$

$$= \Delta_m^{-2} \sum_{j=1}^m \int_{t_j}^{t_{j+1}} \sum_{i=1}^m \left( \int_{t_j}^{t_{j+1}} \langle e_i, u^m(t) - u^m(s)\rangle \, \mathrm{d}s \right)^2 \mathrm{d}t. \quad (32)$$

Using that for $t, s \in [0, T]$ and $i \in \mathbb{N}$ it holds

$$\langle e_i, u^m(t) - u^m(s)\rangle = \langle e_i, u^m(t) - u(t)\rangle + \langle e_i, u(t) - u(s)\rangle + \langle e_i, u(s) - u^m(s)\rangle$$

we can estimate (32), employing the scalar Hölder inequality by

$$3\Delta_m^{-2} \sum_{j=1}^m \int_{t_j}^{t_{j+1}} \sum_{i=1}^m \left( \int_{t_j}^{t_{j+1}} \langle e_i, u^m(t) - u(t)\rangle \, \mathrm{d}s \right)^2 \mathrm{d}t$$

$$+ 3\Delta_m^{-2} \sum_{j=1}^m \int_{t_j}^{t_{j+1}} \sum_{i=1}^m \left( \int_{t_j}^{t_{j+1}} \langle e_i, u(t) - u(s)\rangle \, \mathrm{d}s \right)^2 \mathrm{d}t$$

$$+ 3\Delta_m^{-2} \sum_{j=1}^m \int_{t_j}^{t_{j+1}} \sum_{i=1}^m \left( \int_{t_j}^{t_{j+1}} \langle e_i, u(s) - u^m(s)\rangle \, \mathrm{d}s \right)^2 \mathrm{d}t. \quad (33)$$

The third summand in (33), omitting the constant factor, can be estimated by Hölder's inequality regarding temporal integration in $s$ by

$$\Delta_m^{-1} \sum_{j=1}^m \int_{t_j}^{t_{j+1}} \sum_{i=1}^m \int_{t_j}^{t_{j+1}} |\langle e_i, u^m(s) - u^m(s)\rangle|^2 \, \mathrm{d}s \, \mathrm{d}t,$$

which by Bessel's inequality and $\Delta_m^{-1} \int_{t_j}^{t_{j+1}} \mathrm{d}t = 1$ is bounded by

$$\sum_{j=1}^m \int_{t_j}^{t_{j+1}} \|u(s) - u^m(s)\|^2_{L^2(\Omega)} \, \mathrm{d}s = \|u^m - u\|^2_{\mathcal{Y}} \quad (34)$$

and converges to zero as $m \to \infty$ due to $\mathcal{V} \hookrightarrow\!\!\!\!\to \mathcal{Y}$. Convergence of the first summand in (33) to zero can be argued analogously. It remains to show that

$$\lim_{m\to\infty} \Delta_m^{-2} \sum_{j=1}^m \int_{t_j}^{t_{j+1}} \sum_{i=1}^m \left( \int_{t_j}^{t_{j+1}} \langle e_i, u(t) - u(s) \rangle \, \mathrm{d}s \right)^2 \mathrm{d}t = 0.$$

Applying the integration-by-parts formula of (Roubíček, 2012, Lemma 7.3) with $H = V = L^2(\Omega)$ (which is justified since $\mathcal{V}$ attains at least $H^1(\Omega)$ spatial regularity), we infer that,

$$\langle e_i, u(t) - u(s) \rangle = \int_s^t \langle e_i, \partial_t u(z) \rangle \, \mathrm{d}z,$$

for every $i \in \mathbb{N}$. With this, Bessel's and twice Hölder's inequality we derive that the second summand in (33) can be estimated by

$$\Delta_m^{-1} \sum_{j=1}^m \int_{t_j}^{t_{j+1}} \sum_{i=1}^m \int_{t_j}^{t_{j+1}} |\langle e_i, u(t) - u(s) \rangle|^2 \, \mathrm{d}s \, \mathrm{d}t$$

$$\leq \Delta_m^{-1} \sum_{j=1}^m \int_{t_j}^{t_{j+1}} \sum_{i=1}^m \int_{t_j}^{t_{j+1}} |t - s| \int_s^t |\langle e_i, \partial_t u(z) \rangle|^2 \, \mathrm{d}z \, \mathrm{d}s \, \mathrm{d}t$$

$$\leq \Delta_m^{-1} \sum_{j=1}^m \int_{t_j}^{t_{j+1}} \int_{t_j}^{t_{j+1}} |t - s| \int_s^t \|\partial_t u(z)\|^2 \, \mathrm{d}z \, \mathrm{d}s \, \mathrm{d}t.$$

Using that $|t - s| \leq \Delta_m$ we can estimate this term by

$$\left( \sum_{j=1}^m \int_{t_j}^{t_{j+1}} \int_{t_j}^{t_{j+1}} \mathrm{d}s \, \mathrm{d}t \right) \|\partial_t u\|_{\mathcal{Y}}^2 = \Delta_m T \|\partial_t u\|_{\mathcal{Y}}^2$$

which converges to zero as $m \to \infty$ (since $\Delta_m = T/m$ does). This finally concludes the regularity property (5) for $K^\dagger$ as in (29) and the reduced measurement operators $K^m$ as in (30). $\qquad\square$

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
