# OpenReview forum: "Symbolic Recovery of PDEs from Measurement Data"
_TMLR — Accepted by TMLR_

### Review · Reviewer_cFRf · 2026-02-27

**Summary Of Contributions:**

This paper addresses the problem of recovering Partial Differential Equations (PDEs) from noisy observations. To achieve this, the authors introduce "Symbolic Networks" (SN), a method aimed at recovering PDEs in a symbolically interpretable format. In the SN architecture, activation functions are pre-defined (e.g., trigonometric or exponential functions), and the intermediate layers consist of polynomials with trainable coefficients. Overall, the paper has a strong theoretical focus but offers limited discussion on practical implications.

(Note: My expertise aligns more with applied and empirical machine learning. Therefore, I approach this review primarily from a practical perspective and have not deeply verified the theoretical proofs.)

**Audience:**

No

**Audience Explanation:**

The problem in itsself is high importance. However, I am not sure how relevant the paper is in its current form due to the limited validation of the method.

**Claims And Evidence:**

Yes

**Claims Explanation:**

Identifiability Guarantee: The main contribution of the paper is its rigorous identifiability result. The authors successfully demonstrate that, in the limit of noiseless and complete measurements, Symbolic Networks can uniquely reconstruct the simplest physical law within the PDE model.

**Requested Changes:**

Limited Experimental Validation: Despite the strong theoretical claims, the experimental section lacks practical relevance and scope. The evaluation is restricted to a single, simple one-dimensional PDE where the law depends only on the first derivative of the state. This makes it difficult to assess the real-world utility of the proposed method.

Rigid Architectural Assumptions: The approach heavily relies on the assumption that the underlying physics can be captured by the pre-defined activation functions. The paper does not adequately address what happens when the true physical law falls outside the capacity of this fixed architecture.

---

> ### Author Response · Authors · 2026-04-03
> **Responses to the Comments**
>
> We thank the reviewer for the useful comments, which we address one by one.
>
> **Numerical experiments.** We acknowledge the reviewer’s concern about the limited empirical validation in the original version, a point also raised by the other two reviewers. In response, we have completely rewritten and substantially expanded the experimental section:
> It now includes numerical results for a set of seven physically motivated ODE systems from a standard test dataset, covering a range of difficulties suitable for symbolic model learning, as well as two different two-dimensional PDE systems.
> While our primary focus remains on the theoretical reconstructability results in function space, the expanded experiments are designed to go beyond a mere proof-of-concept and to better support the practical relevance of the framework. However, we emphasize that our main contribution is not a novel neural network architecture for improved numerical performance, but rather an analysis-guided framework to deal with non-uniqueness in symbolic PDE recovery that is applicable to existing architectures (EQL and ParFam). Thus, the main focus of our experiments is still on practical feasability rather than a comprehensive validation of performance improvements.
>
> **Rigid architectural assumption.** We thank the reviewer for pointing this out. Indeed, when writing the paper, we had discussions about whether to include also this second setting of physical laws that fall outside the neural network capacity. After some considerations, however, we decided to address this only partially by providing a roadmap for extending our results in this direction. This was previously discussed after Corollary 14 (original version), and is now provided in an extended version in Subsection 4.3. The two main reasons choosing this path are as follows: 1) While, as outlined in Subsection 4.3, one could adapt all-at-once PDE identification approaches of previous works - Aarset et al. (2023), Holler \& Morina (2024a), Kaltenbacher (2016), Kaltenbacher \& Nguyen (2022) - to address physical laws outside the neural network capacity in the symbolic regression setting (provided suitable approximation properties of the architecture and appropriate regularization are ensured), carrying this out in detail would be highly non-trivial and substantially increase the length of the theoretical part and technical depth of the paper, thereby hiding the setting that is actually in the focus of our work.	2) From a practical perspective, we note that one typically fixes a sufficiently rich architecture in applications (so that the target law is representable for large enough models), and this is also the regime assumed in pioneering NN-based symbolic regression work such as the EQL papers by Lampert et al., where non-representability leads to poor extrapolation (for a cart-pole model) unless the architecture is enlarged.

---

> > ### Comment · Reviewer_cFRf · 2026-04-03
> > **Comment**
> >
> > Thank you for the extensions and detailed answers. I believe the paper significantly improved.

---

### Review · Reviewer_KkaF · 2026-03-16

**Summary Of Contributions:**

This paper studies recovery of PDE right-hand sides from indirect, noisy measurements in an all-at-once inverse-problem framework. The unknown physical law is parameterized by a symbolic network built from rational functions and user-specified base functions, and the paper develops the corresponding function-space analysis. The main theoretical result shows that, under representability and regularity assumptions, minimizers recover the state and converge to an admissible symbolic law with minimal regularization; exact recovery of the true law requires an additional identifiability assumption. The numerical section instantiates the framework with ParFam on two synthetic 1D examples.

The topic is interesting and the mathematical treatment is serious. My main reservations are that the paper sometimes states its conclusions more strongly than what is actually proved, and that the empirical section remains quite limited relative to the scope of the claims.

**Additional Comments:**

I like the general direction of the paper, and I think there is a worthwhile idea here. My hesitation is mostly about the gap between what is mathematically shown, what is empirically demonstrated, and how strongly the paper is written. With a more careful statement of scope, and a significantly stronger experimental section, this could become a solid TMLR submission. In its current form, however, I would not support acceptance.

**Audience:**

Yes

**Audience Explanation:**

The paper sits at a natural intersection of symbolic regression, inverse problems, and scientific machine learning, and I expect that part of the TMLR audience would be interested in this direction. In particular, the function-space treatment of symbolic PDE identification is conceptually worthwhile even though I do not think the paper is ready in its current form.

**Broader Impact Concerns:**

I do not have a broader-impact concern.

**Claims And Evidence:**

No

**Claims Explanation:**

The theory is nontrivial and, at a high level, the paper does support its main mathematical development. However, I do not think the current evidence fully supports the way some of the claims are phrased.

1. The main recovery statement is more qualified than the abstract suggests. Theorem 13 gives subsequential convergence to a regularization-minimizing admissible symbolic law, while exact recovery of the true physical law is only obtained under the additional identifiability condition (I) in Corollary 14. I therefore think the paper should be more careful when describing what is “uniquely reconstructed.”
2. The empirical evidence is still narrow. The numerical section is restricted to two synthetic one-dimensional PDEs in a controlled setup, with simulated multiplicative noise and no real-data experiment. I also did not see baseline comparisons or reporting over multiple random seeds. In its current form, the experimental section reads as an encouraging proof of concept rather than a convincing demonstration of broader practical applicability.

**Requested Changes:**

1. Critical. Tone down the main claims so that they match the actual statements proved in the paper. In particular, distinguish clearly between recovery of an admissible regularization-minimizing law and exact recovery of the true law under assumption (I).
2. Critical. Strengthen the experimental section substantially. At minimum, I would want broader empirical coverage than two simple 1D synthetic examples, together with stronger evidence on robustness. If the paper intends to make claims beyond a controlled proof-of-concept setting, it should either include more demanding PDEs / higher-dimensional settings or narrow the practical claims accordingly.
3. Non-critical but important. The paper would benefit from a sharper positioning of its novelty. As written, part of the contribution reads as an adaptation of an existing all-at-once identification framework to a symbolic-network parameterization. I would like the authors to state more plainly what is genuinely new here relative to the combination of prior all-at-once PDE identification work and existing symbolic regression architectures such as ParFam.

---

> ### Author Response · Authors · 2026-04-03
> **Responses to the Comments**
>
> We thank the reviewer for the review and for the useful comments. We address them one by one.
>
> **Alignment of claims.**  We agree that the passages referring to our recovery results, in particular the one in the abstract, should be made more precise. We now explicitly distinguish between (i) convergence to a regularization-minimizing admissible symbolic law (Theorem 13; now Theorem 9) and (ii) exact recovery of the true physical law, which additionally relies on the identifiability condition (I) (Corollary 14; now Corollary 10). To avoid ambiguity, we have revised the wording throughout the manuscript accordingly, to reflect this distinction more precisely. These clarifications have been incorporated in the Abstract, in the introduction in Subection 1.1 (“Scope and contributions”), the conclusion, and the closing paragraph of Subsection 4.2, where we again review the statements of the main reconstruction results.
>
> **Numerical experiments.** Even though our paper is focused on analysis and the numerical setup was mainly intended to illustrate the analytic identification result,  we see that this was a concern shared by all three reviewers. As consequence, we have completely rewritten and substantially extended the experimental section: We now include numerical results for a set of seven physically motivated ODE systems from a standard test dataset, covering a range of difficulties suitable for symbolic model learning, as well as two different two-dimensional PDE systems. We also clarify that multiple random seeds were already used in our experiments and now explicitly visualize this by reporting median performance with interquartile ranges, so the variability across runs is clear.
> While our primary focus remains on the theoretical reconstructability results in function space, the expanded experiments are designed to go beyond a mere proof-of-concept and to better support the practical relevance of the framework. However, we emphasize that our main contribution is not a novel neural network architecture for improved numerical performance, but rather an analysis-guided framework to deal with non-uniqueness in symbolic PDE recovery that is applicable to existing architectures (EQL and ParFam).
>
> **Clarification of contributions.** We thank the reviewer for pointing this out. In the revised manuscript, we have updated the dedicated subsection on “Scope and contributions” (Subsection 1.1) and added a subsection on "Novelty" (Subsection 1.2) in the Introduction that explicitly contrasts our work to prior all-at-once PDE identification work and neural network based symbolic regression techniques, emphasizing the novelty.

---

> > ### Comment · Reviewer_KkaF · 2026-04-05
> >
> > Thank you for the detailed response and substantial revisions. The paper has improved considerably, and I now consider it a strong and suitable submission for TMLR.

---

### Review · Reviewer_ATa9 · 2026-03-24

**Summary Of Contributions:**

### Summary
This paper concentrates on reconstruction of symbolic expressions and state for partial differential equations (PDEs) from noisy and incomplete measured data. To achieve this, it proposes a framework based on symbolic networks, whose trainable components are rational functions and activations are user-defined base functions. It employs the all-at-once formulation in function space where the state and model parameters are jointly optimized. Theoretical results demonstrate the convergence ability under several assumptions, and numerical experiments further support the theoretical findings and ability of reconstructing physical laws.
### Strengths
1. The problem addressed is important and highly relevant to scientific machine learning
2. The paper provides a rigorous theoretical treatment of symbolic PDE recovery
### Weaknesses
1. The Introduction is overly long and somewhat overloaded with related work. Section 1 contains an extensive literature review, which makes the introduction difficult to read and obscures the main motivation and contributions of the paper. I would suggest shortening this section substantially and moving part of the related work discussion to a separate section. The introduction would be more effective if it focused primarily on the background, motivation, and the main contributions.
2. The theoretical results rely on several assumptions regarding the measurements, the solution regularity, and the regularization framework. While such assumptions may be necessary for the analysis, they seem restrictive from an applied perspective. It would strengthen the paper if the authors could discuss these assumptions in more detail in relation to realistic application scenarios. For example, why they are important, and what may happen when they are violated.
2. The numerical evaluation is relatively limited. The experimental section only considers simple one-dimensional PDE examples, which makes it difficult to assess the broader practical effectiveness of the method. In addition, the paper does not include comparisons with other symbolic PDE discovery approaches or related baselines. The empirical section would be significantly strengthened by including more challenging PDEs, potentially in higher-dimensional or more realistic settings, as well as comparisons with representative competing methods.

**Audience:**

Yes

**Audience Explanation:**

Audience with research interest in machine learning and scientific computing would be interested in this paper.

**Claims And Evidence:**

Yes

**Claims Explanation:**

This paper provides both theoretical and numerical results to support the claims.

**Requested Changes:**

Please the weakness, which provides the advised changes.

---

> ### Author Response · Authors · 2026-04-03
> **Responses to the Comments**
>
> We thank the reviewer for the helpful comments. We address them one by one.
>
> **Structure of introduction.** We thank the reviewer for this suggestion concerning the structure of the introduction. In the revised manuscript, we have separated the original introductory material into Section 1 (“Introduction”), which now focuses on motivation and a concise statement of our main contributions, and Section 2 (“Related work”), which contains the more detailed literature discussion.
>
> **Discussion of assumptions.** We agree that our assumptions on measurements, regularity and regularization merit a more explicit discussion. In response, we have added a dedicated “Discussion of assumptions” (Subsubsection 4.1.2) in which we explain the roles of Assumptions 4 and 7, and comment on their limitations.
>
> **Numerical experiments.** We appreciate the reviewer’s request for a broader and more detailed experimental evaluation, which was also addressed by the other reviewers. In the revised manuscript, we have thoroughly rewritten and extended the numerical section: We now consider a set of seven physically motivated ODE systems of varying difficulty tailored to symbolic model learning, as well as two different two-dimensional PDE systems.
>
> Regarding a comparison with other competing methods, please note that the contribution of our work is not a novel neural network architecture for improved practical performance, but rather an analysis-guided framework to deal with non-uniqueness in symbolic PDE recovery that is applicable to existing architectures (EQL and ParFam). In particular, our framework allows for applicability of these methods also in the setting of indirect, incomplete measurements, a situation that has not yet been considered in other works. Thus, we believe the numerical experiment should focus on practical feasability of our approach using an existing architecture.

---

> > ### Comment · Reviewer_ATa9 · 2026-04-07
> >
> > Thanks for the detailed response and addressing my comments in the revised manuscript. I am satisfied with the revisions, and in my opinion the paper is suitable for publication in TMLR.

---

### Comment · Action_Editor_MsnX · 2026-03-28
**Discussion**

Dear all,

Three reviews are now in and have been made public to authors and all reviewers.
The discussion phase is now ongoing, meaning that authors are expected to reply to the reviews individually and revise the submission accordingly. The reviewers are also encouraged to engage in discussion among each other.

Best,
Action Editor

---

### Author Response · Authors · 2026-04-03
**Thank you to Reviewers and Action Editor**

We thank the reviewers for the time and effort they have invested in providing constructive, and valuable feedback, and we are grateful to the Action Editor for carefully managing the review process. The comments have significantly improved the quality and clarity of our work. In the revised manuscript, we have substantially expanded the section on numerical experiments. Beyond the extended experiments, we have addressed all other points raised by the reviewers. All changes are documented in the main revision, and we have also submitted a separate difference file with the modifications clearly highlighted for convenience. For further details, including point-by-point explanations of how each comment has been addressed, please see the individual responses to the reviewers below.

We hope that the revised manuscript and the expanded experiments satisfactorily address all concerns, and we are happy to clarify any remaining issues.

---

### Author Response · Authors · 2026-04-07
**Final Response to Reviewers**

We thank the reviewers for their positive assessment of our revision. We appreciate that all questions and concerns have been satisfactorily addressed, and that the reviewers now unanimously consider the paper suitable for publication in TMLR.

---

> ### Comment · Action_Editor_MsnX · 2026-04-07
> **Awaiting the "official recommendation" button**
>
> Dear authors and reviewers,
>
> Thanks for the swift replies and the smooth discussion process.
> As the openreview system is currently set up for TMLR, we will have to wait for the "official recommendation" button to show up for reviewers after a predefined period. I can only recommend a decision for the submission, once all reviewers have submitted their official recommendation. Reviewers will receive an email when this is possible.
>
> Best,
> AE

---

### Decision · Action_Editor_MsnX · 2026-04-21

**Recommendation:** Accept as is

**Audience:**

Yes

**Audience Explanation:**

The submission targets a timely topic that has received substantial interest from the broader ML community and is therefore certainly of interest to sizable subsets of  TMLR's audience.

**Claims And Evidence:**

Yes

**Claims Explanation:**

Remaining concerns around the clarity and claims made in the submission have been fully resolved during the discussion period and all reviewers now agree that claims are appropriately supported.